# COME: Test-time adaption by Conservatively Minimizing Entropy

**Qingyang Zhang[1],\*, Yatao Bian[2] [†], Xinke Kong[1], Peilin Zhao[2] and Changqing Zhang[1][†]**
College of Intelligence and Computing, Tianjin University[1]
Tencent AI Lab[2]

## Abstract

Machine learning models must continuously self-adjust themselves for novel data distribution in the open world. As the predominant principle, entropy minimization (EM) has been proven to be a simple yet effective cornerstone in existing test-time adaption (TTA) methods. While unfortunately its fatal limitation (i.e., overconfidence) tends to result in model collapse. For this issue, we propose to **Co**nservatively **M**inimize the **E**ntropy (COME), which is a simple drop-in replacement of traditional EM to elegantly address the limitation. In essence, COME explicitly models the uncertainty by characterizing a Dirichlet prior distribution over model predictions during TTA. By doing so, COME naturally regularizes the model to favor conservative confidence on unreliable samples. Theoretically, we provide a preliminary analysis to reveal the ability of COME in enhancing the optimization stability by introducing a data-adaptive lower bound on the entropy. Empirically, our method achieves state-of-the-art performance on commonly used benchmarks, showing significant improvements in terms of classification accuracy and uncertainty estimation under various settings including standard, life-long and open-world TTA, i.e., up to $34.5\%$ improvement on accuracy and $15.1\%$ on false positive rate. Our code is available at: https://github.com/BlueWhaleLab/COME.

## 1 Introduction

Endowing machine learning models with self-adjust ability is essential for their deployment in the open world, such as autonomous vehicle control and embodied AI systems. To this end, test-time adaption (TTA) emerges as a promising strategy to enhance the performance in the open world which often encounters unexpected noise or corruption (e.g., data from rainy or snowy weather). Unsupervised losses play a crucial role in model adaptation, which can improve the accuracy of a model on novel distributional test data without the need for additional labeled training data. The representative strategy entropy minimization (EM) adapts classifiers by iteratively increasing the probabilities assigned to the most likely classes, and is an integral part in the state-of-the-art TTA methods (Press et al., 2024; Wang et al., 2021; Zhang et al., 2022; Niu et al., 2022; Wang et al., 2022b; Iwasawa & Matsuo, 2021; Niu et al., 2023; Yang et al., 2024). The initial intuition behind using entropy minimization, given by (Wang et al., 2021) is based on the observation that models tend to be more accurate on samples for which they make predictions with higher confidence. The natural extension of this observation is to encourage models to bolster the confidence on test samples.

However, this intuition may not always be true since there always exists irreducible uncertainty which arises from the natural complexity of the data or abnormal outliers. Naturally, one might expect a machine learning model to adapt itself to test data and favor higher confidence on right prediction, but of course not absolute certainty for the erroneous. This contradiction challenges the suitability of EM in TTA tasks, which greedily pursues low-entropy on all test samples. A notable example in recent research concerns that EM can be highly unstable and frequently lead to model collapse when the models encounter unreliable samples in the wild (Niu et al., 2023). In this work, we hypothesize

---

\*Work done during an internship at Tencent AI Lab.
†Co-supervised. Correspondence to Yatao Bian <yatao.bian@gmail.com> and Changqing Zhang <zhangchangqing@tju.edu.cn>.

that due to the nature of EM, previous TTA methods tend to be highly overconfident ignoring the reliability of various test samples, which further results in the unsatisfactory performance.

For the above issues, we propose a simple yet effective model-agnostic learning principle, termed **Co**nservatively **M**inimizing **E**ntropy (`COME`) to stabilize TTA. We first consider the model output as *opinion* which explicitly models the uncertainty of each sample from a Theory of Evidence perspective. Then, we encourage the model to favor definitive opinions for TTA and meanwhile take the uncertainty information into consideration. This offers two-fold advantages compared to EM learning principle. First, our `COME` leverages subjective logic (Jsang, 2018), which is an off-the-shelf uncertainty tool in Bayesian toolbox to effectively perceive the uncertainty raised upon varying test samples without altering the original model architecture or training strategy. Second, when encountering unreliable outliers, the model is regularized to favor conservative confidence and be able to explicitly express *"I do not know"*, i.e., reject to classify them to any known classes, which meets our expectation on model trustworthiness. Theoretically, our `COME` takes inspiration from Bayesian framework, and can be proved to correspond with a data-adaptive upper bound on the model confidence, which is a desirable property for TTA where the reliability of test samples are often varying from time. The contributions of this work are summarized as follows:

- As a principled alternative beyond entropy minimization, we propose a simple yet effective driven strategy for test-time adaption called Conservatively Minimizing Entropy (`COME`) which improves previous methods by exploring and exploiting the uncertainty.

- We provide theoretical analysis with insight in contrast to EM, the model confidence of our `COME` is provably upper bounded in a data-adaptive manner, which enables TTA methods to focus on reliable samples and conservatively handle abnormal test samples.

- We perform extensive experiments under various settings, including standard, open-world and lifelong TTA, where the proposed `COME` achieves excellent performance in terms of both classification accuracy and uncertainty quantification.

## 2 RELATED WORK

**Test-time adaption** aims to bridge the gaps between source and target domains during test-time without accessing the training-time source data. **Entropy minimization** performs an important role in test-time adaption, which has been integrated as a part of numerous TTA methods (Press et al., 2024; Wang et al., 2021; Niu et al., 2022; Wang et al., 2022b; Iwasawa & Matsuo, 2021; Yang et al., 2024; Chen et al., 2022). However, it has been observed that the performance of EM can be highly sub-optimal and unstable when encounter unreliable environments. To this end, previous works incorporate many strategies including i) Samples selection, which selectively filter out the high-entropy unreliable samples by manually setting an entropy threshold (Iwasawa & Matsuo, 2021; Niu et al., 2023). ii) Constrained optimization, which heuristically enforces that the updated parameters do not diverge too much during adaption. iii) Model recovery, which lively monitor the state of the adapting model and frequently reset it when detecting performance collapse (Niu et al., 2023; Wang et al., 2022b). Although these strategies have shown promising performance, the underlying issues of the EM are still largely unexplored. In contrast, this work aims to handle the inherent issues of EM learning objective and validate the necessity and effectiveness in TTA settings.

**Uncertainty quantification** is one key aspect of the model reliability. With accurate uncertainty estimation ability, further processing can be taken to improve the performance of machine learning systems (e.g., human assistance) when the predictive uncertainty is high. This is especially useful in high-stake scenarios such as medical diagnosis (Wang et al., 2023). To obtain the uncertainty, Bayesian neural networks (BNNs) (Denker & LeCun, 1990; Mackay, 1992) have been proposed to replace the deterministic weight parameters of model with distribution. Unlike BNNs, ensemble-based methods obtain the uncertainty by training multiple models and ensembling them (Rahaman et al., 2021; Abe et al., 2022). In this paper, we focus on estimating and exploiting uncertainty under the theory of subjective logic (SL, (Jsang, 2018)). Unlike BNNs or ensemble, SL explicitly models the uncertainty in a single forward pass without modifying the training strategy or model architecture, which meets our expectation of computational effectiveness for TTA tasks.

## 3 MOTIVATION

We consider the fully test-time adaption setting in $K$-classification task where $\mathcal{X}$ is the input space and $\mathcal{Y} = \{1, 2, ..., K\}$ denotes the target space. Given a classifier $f : \mathcal{X} \to \mathbb{R}^K$ parameterized by $\theta$ which has been pretrained on training distribution $P^{\text{train}}$, our goal is to boost $f$ by updating its parameters $\theta$ online on each batch of test data drawn from test distribution $P^{\text{test}}$. Note that in fully TTA setting, the training data $P^{\text{train}}$ is unavailable and one can only tune $\theta$ on unlabeled test data. This is derived from realistic concerns of privacy, bandwidth or profit. Compared to other closely related setting, i.e., Source-Free Domain Adaptation (SFDA), TTA focuses on online adjusting during the testing while SFDA generally perform offline. That is, TTA method aims on (unsupervised) adaption online and the inference latency matters. **Entropy minimization (EM)** algorithm iteratively optimizes the model to minimize the predictive entropy on test sample $x$

$$H(p(y|x)) = -\sum_{k=1}^{K} p(y = k|x) \log p(y = k|x), \tag{1}$$

where $p(y|x)$ is the class distribution calculated by normalizing the output logits $f(x)$ with softmax function, i.e., $p(y = k|x) = \frac{\exp f_k(x)}{\sum \exp f(x)}$. $H$ is the Shannon's entropy.

**Other learning objectives.** Besides EM, there also exists several TTA methods which explore other unsupervised learning objectives. Notable examples include 1) Pseudo label (PL): $\mathcal{L}_{\text{PL}} = -\mathbb{E} \log p(y = \hat{y}|x)$ which encourages the adapted model to fit the pseudo label $\hat{y}$ predicted by the pretrained model, 2) Module adjustment (T3A) which adjusts the parameters in the last fully connected layer, and can be viewed as an implicit way to minimize entropy (Iwasawa & Matsuo, 2021), 3) Energy minimization: $\mathcal{L}_{\text{TEA}} = -\mathbb{E} \log \sum_{k=1}^{K} \exp f(x)$ which aims to minimize the free energy during adaption, and takes inspiration from energy model (Yige et al., 2024), 4) Contrastive learning objective: $\mathcal{L}_{\text{infoNCE}} = -\log \frac{\exp \text{query} \cdot \text{key}^+}{\sum \exp \text{query} \cdot \text{key}}$ which strives to minimize the cosine distance between the query and positive samples ($\text{key}^+$) while maximizing the cosine distances between query and negative samples (Chen et al., 2022), 5) The recent advanced FOA (Niu et al., 2024) which uses evolution strategy to minimize the test-training statistic discrepancy and model prediction entropy.

**The overconfident issue of EM.** We begin by testing EM in standard TTA setting, and put forward the following observations to detail its unsatisfying performance.

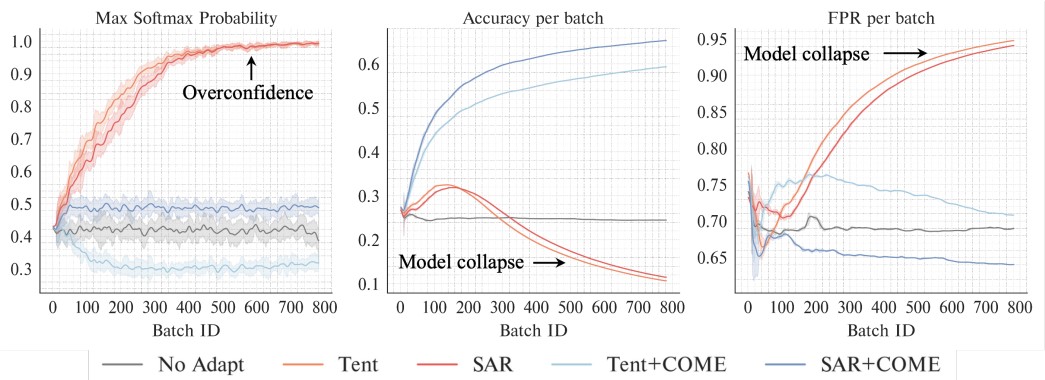

Figure 1: Empirical observations of Entropy Minimization when equipped to two representative TTA methods, i.e., the seminal Tent (Wang et al., 2021) and recent advanced SAR (Niu et al., 2023). Along the TTA process, the uncertainty of models tuned with EM quickly drops, and the false positive rate decreases temporarily for a very short time horizon before quickly increasing. Along the same adaption trajectory, the model accuracy also improves for a short time compared to the initial model and then quickly decreases, after which the model collapses to a trivial solution. By contrast to EM, our COME establishes a stable TTA process with consistently improved classification accuracy and false positive rate. Besides, the model confidence of our COME is much more conservative, which implies fewer risks of overconfidence and a more accurate uncertainty awareness.

As shown in Figure 1, EM tends to overconfident prediction and assign extremely high probability to one certain class. The absolutely high confidence on all test samples is obviously a rather undesired characteristic. We test on ImageNet-C under snow corruption of severity level 5 as a typical showcase, and refer interested readers to Appendix C.11 for more similar results.

## 4 METHODOLOGY

We propose to conservatively minimize the entropy under uncertainty modeling, a simple alternative to EM algorithm. The key idea of COME is to quantify and then regularize the uncertainty during TTA without altering the model architecture or training strategy, which avoids the overconfident nature of EM at minimal cost. We first introduce uncertainty quantification by the subjective logic and then present how to regularize the uncertainty during TTA.

### 4.1 MODELING UNCERTAINTY BY THE SUBJECTIVE LOGIC

Given a well trained classifier $f : \mathcal{X} \rightarrow \mathcal{Y}$, the most simple way to quantify the uncertainty of each sample is using the softmax probability as confidence in prediction. A few pioneer works propose to filter out the test samples with high-entropy predicted softmax probability for stable TTA (Niu et al., 2023; 2022). However, it has been shown that softmax probability often leads to overconfident predictions, even when the predictions are wrong or the inputs are abnormal outliers (Moon et al., 2020; Van Amersfoort et al., 2020). Thus this simple strategy may not be satisfied enough and highlights the necessity of better uncertainty modeling. To this end, we propose to obtain the uncertainty through subjective logic, which defines a framework for obtaining the probabilities (belief masses) of different classes and the overall uncertainty (uncertainty mass) based on the *evidence* [1] collected from data. Specifically, in $K$ classification task, SL formalizes the belief assignments over a frame of discernment as a Dirichlet distribution. In contrast to softmax function that directly normalizes the logits $f(x)$ to model the class distribution $p(y|x)$, SL considers the model output as evidence (denoted as $e$) to model a Dirichlet distribution which represents the density of all possible probability assignment $\boldsymbol{\mu} = [p(y = 1|x), p(y = 2|x), \ldots, p(y = K|x)]$. That is, the predicted categoricals $\boldsymbol{\mu}$ is also a random variable itself, which yields a Dirichlet distribution as follow

$$p(\boldsymbol{\mu}|x) = \text{Dir}(\boldsymbol{\mu}|\boldsymbol{\alpha}) = \frac{1}{B(\boldsymbol{\alpha})} \prod_{k=1}^{K} \mu_k^{\alpha_k - 1}, \ \boldsymbol{\alpha} = \boldsymbol{e} + 1, \tag{2}$$

where $\text{Dir}(\boldsymbol{\mu}|\boldsymbol{\alpha})$ is the Dirichlet distribution characterized by parameters $\boldsymbol{\alpha}$. The summation of all $\alpha_k \in \boldsymbol{\alpha}$ is so called the strength $S$ of the Dirichlet distribution, i.e., $S = \sum_k \alpha_k = \sum_k e_k + 1$. Then SL tries to assign a belief mass $b_k$ to each class label $k$ and an overall uncertainty mass $u$ to the whole frame based on the collected evidence as follow

$$b_k = \frac{e_k}{S} = \frac{\alpha_k - 1}{S} \text{ and } u = \frac{K}{S}, \text{ subject to } u + \sum_{k=1}^{K} b_k = 1, \tag{3}$$

where $S$ is the Dirichlet strength which denotes the total evidence we collected and $K$ is the total classes number. Eq. 3 actually describes the phenomenon where the more evidence observed for the $k$-th category, the greater the belief mass assigned to the $k$-th class. Correspondingly, the less total evidence $S$ observed, the greater the total uncertainty $u$. Such assignment is so called the subjective opinion

$$\mathcal{M}(x) = [b_1, b_2, \cdots, b_k, u], \tag{4}$$

which not only describes the belief of assigning $x$ to each class $k$ but also explicitly models the uncertainty due to lack-of-evidence. To ensure that $\alpha \geq 1$ which meets the requirements of valid parameters of a Dirichlet, we first deploy ReLU function on the logits and then use exponential output function for obtaining the parameters of Dirichlet distribution from the model output $f(x)$ (i.e., logits), where $\boldsymbol{\alpha} = \exp(\text{ReLU}(f(x)))$ and $\boldsymbol{e} = \boldsymbol{\alpha} - 1$.

**Design choices of transformation function.** By definitions, the parameters of a Dirichlet distribution $\boldsymbol{\alpha}$ must be greater than 1 and the evidence $\boldsymbol{e}$ should be non-negative. This can be achieved by applying ReLU activation function or exponential function to the output logits as suggested in

---

[1] In Bayesian context, evidence refers to the metrics collected from the input to support the classification.

previous works (Han et al., 2022; Malinin & Gales, 2018). Alternatively, we can get the evidence by $e = \text{ReLU}(f(x))$ or $e = \exp f(x) - 1$. In this paper, exponential function is chosen since we assume the model is pretrained with standard cross-entropy loss. Thus using exponential function to get the evidence can keep the training strategy unchanged. We refer interested readers to (Malinin & Gales, 2018) and Gal's PhD Thesis (Gal et al., 2016) for more detailed implementation instructions and math deviations.

**Benefits of modeling uncertainty based on subjective opinion for TTA.** It has been widely recognized that using the softmax output as confidence often leads to overconfidence phenomenon (Guo et al., 2017; Hendrycks & Gimpel, 2017; Liu et al., 2020). Other advanced uncertainty measurement methods such as MC dropout (Gal & Ghahramani, 2016), Bayesian neural networks (Huang et al., 2022), deep ensemble (Lakshminarayanan et al., 2017; Rahaman et al., 2021), calibration (Guo et al., 2017; Han et al., 2024) usually require additional computations during inference or a separate validation set, which are not applicable in an unsupervised TTA task. By contrast, the introduced SL directly deduces an additional uncertainty mass through one single forward pass, which is model-agnostic, light-weight, and can be seamlessly integrated in standard pretrained classifier. We defer empirical comparisons between the proposed method and other Bayesian methods or learning objectives that originate from the adaption of the unsupervised domain to Table 10 in the Appendix.

## 4.2 MODEL ADAPTION BY SHARPENING THE OPINION

Vanilla EM minimizes the softmax entropy of the predicted class distribution $p(y|x)$, which inevitably results in assigning rather high probability to one certain class. In contrast, we propose the learning principle to minimize the entropy of opinion

$$\underset{\theta}{\text{minimize}}\ H(\mathcal{M}(x)) = -\sum_{k=1}^{K} b_k \log b_k - u \log u. \tag{5}$$

Compared to entropy minimization on softmax probability, which ultimately assigns all the probability to one certain class, the above learning principle offers the model with an additional option, i.e., express high overall uncertainty and reject to adapt when the observed total evidence is insufficient. In other words, by assigning all belief masses (probability) to uncertainty $u$, the model can now express "*I do not know*" as its predicted opinion. Alternatively, an additional hyperparameter $\lambda$ can be introduced to weight the last term as $-\sum_{k=1}^{K} b_k \log b_k - \lambda u \log u$. The magnitude reflects our confidence in whether the test sample should be adapted by the model or not. We validate the effectiveness of this generalization to Table 12 in the Appendix.

## 4.3 REGULARIZING UNCERTAINTY IN AN UNSUPERVISED MANNER

While subjective logic offers an opportunity to model uncertainty and reject to classify unreliable samples, naively minimizing the entropy of opinion for TTA may still be problematic. As shown in previous works, the model pretrained with softmax output function frequently suffers from overconfidence issue (Guo et al., 2017; Nguyen et al., 2015; Hendrycks et al., 2019). Therefore, the belief mass assigned to the one certain class $k$ by the pretrained model is usually much larger than the uncertainty mass $u$. This results in the model tendency of increasing the belief mass during the entropy minimization process, while neglecting the uncertainty function $u$. Motivated by the above analysis, our next goal is to devise an effective regularization strategy for the uncertainty mass. In supervised learning tasks, previous works leverage labeled training data to constrain the uncertainty mass (Sensoy et al., 2018; Malinin & Gales, 2018). However, these strategies is not applicable due to the unsupervised nature of TTA task where the training data is unavailable. This motivates us to explore the uncertainty information lies in the pretrained model itself for regularization without additional supervision. As one of the simplest yet effective design choices, we propose to constrain the uncertainty mass predicted by the adapted model not to diverge too far from the pretrained model. This results in the following constrained optimization objective

$$\underset{\theta}{\text{minimize}}\ H(\mathcal{M}(x))\ \text{subject to}\ |u_\theta(x) - u_{\theta_0}(x)| \le \delta, \tag{6}$$

where $\theta, \theta_0$ denote the adapted and pretrained model respectively, and $u$ is the uncertainty estimated by Eq. 3 and $\delta$ is a threshold to prevent overly extreme model uncertainty. The magnitude of $\delta$ represents our tolerance for uncertainty divergence.

**Rethink why we constrain on** $|u(x) - u_0(x)| \leq \delta$**.** The uncertainty estimated by pretrained model may not be ideal. However, since in fully TTA task, we can only access unlabeled test data coming online, and the inference efficiency matters. Thus traditional methods devised for handling overconfidence like calibration, ensembling, and other Bayesian methods are not applicable. The only practically available choice is to explore the uncertainty information contained within the model itself. As shown in previous works (Sensoy et al., 2018; Malinin & Gales, 2018), while the softmax probability of pretrained model tend to be overconfident, subjective logic is much more reliable, which can support the proposed regularization. The most straightforward way to realize the aforementioned constraint is to storage the model before and after adaption and explicitly compare the uncertainty mass. However, this may violate the efficiency requirements of TTA. Thus considering the difficulty of constrained optimization in modern neural networks, our next target is to find a way to convert Eq. 6 into an unconstrained form. To this end, we introduce the following Lemma.

**Lemma 1.** *For any $x \in \mathcal{X}$, we have*

$$\frac{K}{||f(x)||_p + \log K} \leq u \leq \frac{K^{1+1/p}}{||f(x)||_p}, \tag{7}$$

*where $f(x)$ is the model output logits, $K$ is the total class number and $|| \cdot ||_p$ denotes the $p$-norm.*

Lemma 1 shows that the uncertainty mass of subjective opinion is bounded by the norm of the total evidence collected from the model output. Thus instead of directly constraining on $u(x)$, we can alternatively constrain on the $p$-norm of model output logits, which is more flexible. Taking inspiration from previous work in supervised learning literature (Wei et al., 2022), this can be achieved by factorize $f(x)$ into $f(x)/||f(x)||_p \cdot ||f(x)||_p$ and then enforcing the gradient on the second term to be equal to zero during optimization. Specifically, the final minimizing objective of COME is

$$\underset{\theta}{\text{minimize}} \; H(\mathcal{M}(x)) \;\; \text{where} \; f(x) = \frac{f(x)}{||f(x)||_p} \cdot ||f(x)||_p^{\text{no\_grad}} \cdot \tau, \tag{8}$$

and $||f(x)||_p^{\text{no\_grad}}$ is the $p$-norm of $f(x)$ with zero gradient. This can be achieved by applying the detach operation which is a common used function in modern deep learning toolbox like PyTorch and TensorFlow. By doing so, minimizing the entropy of opinion would not influence $||f(x)||_p$. $\tau$ is a hyper-parameter which controls the magnitude of recovered logits. Our COME can be implemented by modifying only a few lines of code in the original EM algorithm (shown as Algorithm 1).

**How to set $\tau$ and $p$ in practice?** Noted that the tightness for the upper and lower bounds in Lemma 1 is determined by the choice of $p$. The ratio between the upper and lower bound is minimized by $p = \infty$. The strictness of such constraint should be selected per need by the user via trial and error: if users are extremely cautious about unreliable TTA, $p$ should be tuned up; otherwise, if a better performance is required. In our experiments, we choose $p = 2$ and $\tau = 1$ in accordance with Occam's Razor. Experiments on different hypeparameters can be found in Table 13 in the Appendix.

---

**Algorithm 1:** Pseudo code of COME in a PyTorch-like style.

```
# x: the output logits, model: the test model
def entropy_of_opinion(x):
    belief = exp(x) - 1 / sum(exp(x)) # belief mass
    uncertainty = K / sum(exp(x)) # uncertainty mass
    opinion = cat([belief, uncertainty]) # subjective opinion
    return -sum(opinion * log(opinion)) # entropy of opinion

for data in test_loader: # load a minibatch data
    x = model(data) # forward
    x = x / norm(x, p=2) * norm(x, p=2).detach() # constraint in Eq.9
    loss = entropy_of_opinion(x) # calculate loss
    # ... [backwards and update the parameters]
```

---

**Stability of COME.** We provide preliminary theoretical understanding of the superiority of COME. As we mentioned before, one notable limitation of EM is that it enforces low entropy for all test samples while ignores the instinct complexity of wild test data. Thus at the end of TTA progress, EM ultimately produces model that yields overconfident prediction. Our COME resolves this issue and

introduces an upper bound for each test sample $x$ according to its trustworthiness. This property is formalized as follows

**Theorem 1** (Model confidence upper bound). *For any $x \in \mathcal{X}$, if $|u(x) - u_0(x)| \leq \delta$ holds, then we have*

$$\max_k p(y = k|x) \leq \frac{1}{1 + (K-1)\exp\left(-\frac{K}{u_0 - \delta}\right)}, \tag{9}$$

*where $\max_k p(y = k|x)$ is the model confidence (class probability assigned to the most likely class) and $K$ is the total class number. $u_0$ is the shorthand of $u_{\theta_0}(x)$.*

From Theorem 1, we find that the model confidence in COME has a sample-wise upper bound according to $u_0(x)$. In particular, it implies that the model confidence upper bound of the most likely class decreases according to $u_0(x)$. For this reason, one can suspect that if the test model is uncertain about some sample $x$ (with a rather large $u_0$), it will be difficult to further increase the model confidence on such $x$, which is a desirable property for TTA in the wild.

## 5 EXPERIMENTS

We conduct experiments on multiple datasets with distributional shift to answer the following questions. Q1. In the standard TTA setting, does the proposed method outperform other algorithms? Q2. How does COME perform in more realistic TTA settings, such as open-world TTA or lifelong TTA? Q3. Uncertainty quantification is both the motivation behind COME and the reason for its effectiveness, does our method achieves more reliable uncertainty estimation during TTA? Q4. Ablation study - what is the key factor of performance improvement in our method? Q5. Hyperparameters - how is the performance of our method under different parameters?

### 5.1 SETUP

**Datasets**. Following the common practice (Niu et al., 2022; 2023), we conduct experiments on standard covariate-shifted distribution datasets ImageNet-C (a large-scale benchmark with 15 types of diverse corruption), ImageNet-R and ImageNet-S. Besides, we also consider open-world test-time adaption setting, where the test data distribution $P^{\text{test}}$ is a mixture of both normal covariate-shifted data $P^{\text{Cov}}$ and abnormal outliers $P^{\text{Outlier}}$ of which the true labels do not belong to any known classes in $P^{\text{train}}$. Following previous work in open-set OOD generalization literature (Lee et al., 2023; Bai et al., 2023; Baek et al., 2024), $P^{\text{Outlier}}$ is a suit of diverse datasets introduced by (Yang et al., 2022), including iNaturalist, Open-Image, NINCO and SSB-Hard. **Compared methods.** We compare our COME with a board line of test-time adaption methods, including both EM-based and non-EM based TTA methods. ∘ EM-based methods choose entropy minimization as their learning objective, including Tent (Wang et al., 2021), EATA (Niu et al., 2022), CoTTA (Wang et al., 2022b), MEMO (Zhang et al., 2022) and recent advanced SAR (Niu et al., 2023). ∘ Non-EM methods employ other learning objectives including Pseudo Label (PL), module adjustment (Iwasawa & Matsuo, 2021) (T3A) and energy minimization (Yige et al., 2024) (TEA). Following (Niu et al., 2023), we use the ViT-base architecture as our backbone and defer the results on ResNet50 to Table 17 in the Appendix. The test batch size is 64. When equipped to previous EM-based TTA baselines, we only replace the learning objective with our COME and keep all the other configures consistent to the official implementation. **Tasks and Metrics.** For classification performance comparison, we report the accuracy (Acc) on covariate-shifted data. Besides, for open-world TTA, we report the false positive rate (FPR). The mis-classified samples and outliers are considered as positive samples which should be of higher uncertainty compared to correct classification that is considered as negative. For all experiments, we run multiple times and report the average. Full results with standard deviation are deferred to the Appendix.

### 5.2 EXPERIMENTAL RESULTS

**Performance comparison in standard TTA settings (Q1).** As shown in Table 1, our COME establishes strong overall performance in terms of both classification accuracy. We highlight a few essential observations. Compared to EM learning principle, our COME consistently outperforms it when equipped to the same baseline methods, including Tent (Wang et al., 2021), EATA (Niu

et al., 2022), CoTTA (Wang et al., 2022b) and SAR (Niu et al., 2023). As an example of our method's improved performance, when equipped to the recent SAR, our method yields an accuracy of $64.2\%$, which outperforms the original implementation based on EM of $10.1\%$. Besides, we also compare to Non-EM TTA methods, including TEA, T3A and PL. These methods do not rely on EM learning objective, yet are less effective than EM in terms of classification performance. For a more comprehensive evaluation, we conduct additional experiments on ImageNet-R and ImageNet-S, the results can be found in Table 2.

Table 1: Classification accuracy comparison on ImageNet-C (level 5). Substantial ($\geq 0.5$) improvement and degradation compared to the baseline are highlighted in blue or brown respectively. The detailed results with standard deviation are defer to Table 8 in Appendix C.1.

| Methods | COME | Noise | | | Blur | | | | Weather | | | | Digital | | | |
|---|---|---|---|---|---|---|---|---|---|---|---|---|---|---|---|---|
| | | Gauss. | Shot | Impul. | Defoc | Glass | Motion | Zoom | Snow | Frost | Fog | Brit. | Contr. | Elast. | Pixel | JPEG |
| No Adapt | ✗ | 35.1 | 32.2 | 35.9 | 31.4 | 25.3 | 39.4 | 31.6 | 24.5 | 30.1 | 54.7 | 64.5 | 49.0 | 34.2 | 53.2 | 56.5 |
| PL | ✗ | 49.9 | 48.8 | 50.9 | 48.5 | 41.2 | 52.7 | 42.7 | 24.6 | 42.5 | 63.6 | 73.1 | 65.3 | 44.1 | 63.9 | 62.6 |
| T3A | ✗ | 34.6 | 31.5 | 35.5 | 32.7 | 27.5 | 40.7 | 33.5 | 25.6 | 30.8 | 56.5 | 64.9 | 50.8 | 38.0 | 54.3 | 58.4 |
| TEA | ✗ | 44.5 | 39.3 | 45.8 | 37.6 | 35.4 | 46.4 | 31.4 | 8.9 | 46.4 | 60.1 | 72.5 | 59.5 | 45.7 | 62.3 | 58.9 |
| LAME | ✗ | 34.8 | 31.9 | 35.5 | 30.9 | 24.4 | 38.9 | 30.7 | 23.4 | 29.5 | 53.3 | 64.2 | 41.0 | 32.7 | 52.8 | 56.0 |
| FOA | ✗ | 46.6 | 43.9 | 48.3 | 47.1 | 40.7 | 49.3 | 43.7 | 53.8 | 52.8 | 64.2 | 76.2 | 63.8 | 48.5 | 62.1 | 64.0 |
| Tent | ✗ | 52.5 | 52.1 | 53.4 | 52.8 | 47.4 | 56.7 | 47.4 | 10.5 | 26.4 | 67.2 | 74.3 | 67.3 | 50.4 | 66.5 | 64.6 |
| | ✓ | 53.9 | 53.9 | 55.2 | 55.8 | 51.8 | 59.8 | 52.6 | 58.4 | 61.3 | 71.3 | 78.2 | 68.8 | 57.9 | 70.5 | 68.2 |
| | Improve | △1.4 | △1.9 | △1.8 | △2.9 | △4.4 | △3.2 | △5.2 | △47.9 | △34.9 | △4.1 | △3.9 | △1.5 | △7.6 | △4.0 | △3.6 |
| EATA | ✗ | 56.0 | 56.1 | 57.1 | 54.5 | 54.8 | 59.6 | 58.7 | 61.8 | 60.1 | 71.4 | 75.3 | 68.5 | 62.7 | 69.0 | 66.5 |
| | ✓ | 56.1 | 56.5 | 57.2 | 58.0 | 57.9 | 62.6 | 59.3 | 65.6 | 63.5 | 72.6 | 78.0 | 69.5 | 66.6 | 72.5 | 70.5 |
| | Improve | △0.1 | △0.3 | △0.1 | △3.4 | △3.1 | △3.1 | △0.6 | △3.9 | △3.4 | △1.3 | △2.7 | △1.0 | △3.9 | △3.5 | △4.0 |
| SAR | ✗ | 51.9 | 51.7 | 52.8 | 51.5 | 48.9 | 55.5 | 49.5 | 22.2 | 46.9 | 66.2 | 72.9 | 65.8 | 50.9 | 64.0 | 62.8 |
| | ✓ | 56.4 | 56.6 | 57.4 | 58.3 | 56.9 | 62.9 | 58.3 | 65.3 | 64.5 | 72.7 | 78.5 | 69.6 | 64.0 | 71.9 | 69.7 |
| | Improve | △4.5 | △4.9 | △4.6 | △6.8 | △8.1 | △7.4 | △8.8 | △43.1 | △17.5 | △6.5 | △5.5 | △3.8 | △13.2 | △7.9 | △6.9 |
| CoTTA | ✗ | 40.3 | 37.6 | 41.7 | 34.3 | 28.3 | 44.0 | 35.6 | 38.0 | 43.0 | 58.8 | 70.3 | 58.4 | 39.8 | 58.1 | 59.9 |
| | ✓ | 43.5 | 40.9 | 45.5 | 36.9 | 29.7 | 48.1 | 37.8 | 40.7 | 42.0 | 62.3 | 73.6 | 58.9 | 42.8 | 63.5 | 63.8 |
| | Improve | △3.1 | △3.4 | △3.8 | △2.6 | △1.4 | △4.0 | △2.2 | △2.7 | ▽1.0 | △3.5 | △3.3 | △0.5 | △3.0 | △5.4 | △3.9 |
| MEMO | ✗ | 39.7 | 36.5 | 39.8 | 32.4 | 25.8 | 40.3 | 34.7 | 27.5 | 32.8 | 53.5 | 66.2 | 56.0 | 35.7 | 55.9 | 58.2 |
| | ✓ | 40.6 | 37.5 | 40.6 | 33.4 | 26.7 | 41.2 | 35.4 | 28.7 | 33.7 | 54.7 | 67.1 | 55.9 | 36.6 | 57.2 | 59.3 |
| | Improve | △0.8 | △1.0 | △0.8 | △1.0 | △0.9 | △1.0 | △0.7 | △1.2 | △0.9 | △1.2 | △0.8 | ▽0.1 | △0.9 | △1.3 | △1.1 |

Table 2: Additional results on ImageNet-R and ImageNet-S.

| Dataset | COME | Tent | EATA | SAR | COTTA |
|---|---|---|---|---|---|
| ImageNet-R | ✗ | $37.73 \pm 0.03$ | $36.11 \pm 0.12$ | $51.90 \pm 0.35$ | $36.04 \pm 0.06$ |
| | ✓ | $39.05 \pm 0.05$ | $38.22 \pm 0.23$ | $56.40 \pm 0.18$ | $37.39 \pm 0.08$ |
| ImageNet-S | ✗ | $31.63 \pm 0.07$ | $36.11 \pm 0.19$ | $33.92 \pm 0.69$ | $30.84 \pm 0.35$ |
| | ✓ | $39.22 \pm 0.04$ | $39.22 \pm 0.32$ | $43.52 \pm 0.52$ | $33.82 \pm 0.73$ |

**Performance comparison in open-world and lifelong TTA settings (Q2).** In Table 3 and 4, we present the results under open-world and lifelong TTA settings respectively. In open-world TTA, the test data distribution is a mixture of both normal covariate-shifted data and abnormal outliers. The mixture ratio of $P^{\text{Cov}}$ and $P^{\text{Outlier}}$ is 0.5 following previous work (Bai et al., 2023), i.e., $P^{\text{test}} = 0.5P^{\text{Cov}} + 0.5P^{\text{Outlier}}$. Such outliers arise from unknown classes that are not present in training data, which should not be of high uncertainty for model trustworthiness. According to the experimental results, it is observed that our COME can consistently improve the performance of existing TTA methods.

**Reliability of uncertainty estimation (Q3).** We visualize the distribution of model confidence, i.e., the maximum predicted class probability[2] in open-world TTA setting, where the covariate-shifted samples is ImageNet-C (Gaussian noise level 3), and outliers are NINCO. As shown in Figure 2, the model confidence of our COME can effectively perceive incorrect predictions, which establishes an distinguishable margin. In contrast to the model confidence of EM which is almost identical and nearly $100\%$ for all test samples, the model confidence of our method can provide more meaningful information with which to differentiate correct and wrong predictions.

---

[2]For our COME, the class probability is calculated according to Eq. 5.

Table 3: Classification and uncertainty estimation comparisons under **open-world** TTA settings, where $P^{\text{test}}$ is a mixture of both covariate-shifted samples (Gaussian noise of severity level 3) and a suit of diverse abnormal outliers. Additional results with standard deviation are in Appendix C.2.

| Method | COME | None Acc↑ | FPR↓ | NINCO Acc↑ | FPR↓ | iNaturist Acc↑ | FPR↓ | SSB-Hard Acc↑ | FPR↓ | Texture Acc↑ | FPR↓ | Places Acc↑ | FPR↓ |
|---|---|---|---|---|---|---|---|---|---|---|---|---|---|
| No Adapt | ✗ | 64.4 | 63.7 | 64.5 | 69.9 | 64.4 | 69.5 | 64.4 | 72.5 | 64.8 | 65.6 | 64.3 | 56.8 |
| PL | ✗ | 69.1 | 62.8 | 65.6 | 71.6 | 68.8 | 69.5 | 68.4 | 75.9 | 66.1 | 66.0 | 66.6 | 59.7 |
| T3A | ✗ | 64.4 | 71.2 | 64.3 | 70.0 | 64.2 | 75.0 | 63.7 | 80.7 | 64.4 | 69.0 | 63.8 | 69.5 |
| TEA | ✗ | 63.9 | 63.8 | 60.5 | 72.5 | 62.3 | 74.6 | 63.3 | 79.4 | 61.0 | 67.8 | 61.9 | 64.5 |
| LAME | ✗ | 64.1 | 64.4 | 64.1 | 72.3 | 64.1 | 72.4 | 64.2 | 74.0 | 64.7 | 68.8 | 64.0 | 61.3 |
| FOA | ✗ | 67.8 | 61.2 | 66.4 | 70.5 | 67.4 | 66.1 | 67.1 | 75.6 | 65.8 | 61.4 | 66.6 | 54.1 |
| Tent | ✗ | 70.8 | 63.2 | 66.2 | 71.9 | 69.9 | 70.7 | 69.8 | 77.4 | 66.4 | 66.5 | 68.3 | 59.9 |
|  | ✓ | 72.6 | 64.7 | 68.9 | 64.3 | 72.5 | 63.7 | 72.7 | 70.7 | 68.4 | 60.4 | 70.7 | 45.9 |
|  | Improve | △1.7 | △1.6 | △2.7 | ▽7.6 | △2.6 | ▽7.0 | △2.9 | ▽6.7 | △2.1 | ▽6.1 | △2.4 | ▽14.0 |
| EATA | ✗ | 70.3 | 63.7 | 66.4 | 68.6 | 70.3 | 71.5 | 70.0 | 77.4 | 67.3 | 67.1 | 68.8 | 61.9 |
|  | ✓ | 73.4 | 62.7 | 70.1 | 60.5 | 73.2 | 63.3 | 73.0 | 70.5 | 70.5 | 55.8 | 72.3 | 45.6 |
|  | Improve | △3.1 | ▽1.0 | △3.7 | ▽8.2 | △2.9 | ▽8.2 | △3.0 | ▽6.9 | △3.2 | ▽11.3 | △3.6 | ▽16.3 |
| SAR | ✗ | 69.7 | 62.3 | 64.9 | 71.4 | 66.9 | 70.9 | 67.7 | 78.1 | 64.4 | 64.5 | 66.1 | 58.6 |
|  | ✓ | 73.1 | 62.9 | 69.8 | 66.3 | 73.2 | 65.2 | 73.5 | 71.8 | 69.5 | 59.4 | 72.3 | 49.8 |
|  | Improve | △3.5 | △0.6 | △4.9 | ▽5.1 | △6.3 | ▽5.6 | △5.9 | ▽6.3 | △5.1 | ▽5.1 | △6.3 | ▽8.8 |
| CoTTA | ✗ | 67.6 | 63.4 | 65.3 | 69.7 | 70.4 | 69.5 | 70.3 | 76.0 | 65.8 | 66.2 | 66.6 | 59.2 |
|  | ✓ | 70.5 | 62.5 | 66.2 | 68.8 | 72.4 | 73.5 | 72.2 | 78.7 | 66.5 | 64.7 | 68.9 | 55.0 |
|  | Improve | △2.9 | ▽0.9 | △0.9 | ▽0.9 | △2.0 | △4.0 | △2.0 | △2.7 | △0.7 | ▽1.5 | △2.3 | ▽4.2 |
| MEMO | ✗ | 64.8 | 69.8 | 64.8 | 77.5 | 64.7 | 71.9 | 64.8 | 77.3 | 65.0 | 79.3 | 64.6 | 71.5 |
|  | ✓ | 65.2 | 67.8 | 65.9 | 76.4 | 65.2 | 70.8 | 65.3 | 75.0 | 65.4 | 76.7 | 65.3 | 67.6 |
|  | Improve | △0.5 | ▽1.9 | △1.1 | ▽1.1 | △0.5 | ▽1.1 | △0.5 | ▽2.3 | △0.4 | ▽2.6 | △0.7 | ▽3.9 |

Table 4: Classification and uncertainty estimation comparisons under **lifelong** TTA settings. The model is online adapted and the parameters will never be reset, yet the test input distribution might exhibit a continual shift over time.

| Methods | COME | Noise Gauss. | Shot | Impul. | Blur Defoc | Glass | Motion | Zoom | Weather Snow | Frost | Fog | Brit. | Digital Contr. | Elast. | Pixel | JPEG |
|---|---|---|---|---|---|---|---|---|---|---|---|---|---|---|---|---|
| No Adapt | ✗ | 35.1 | 32.2 | 35.9 | 31.4 | 25.3 | 39.4 | 31.6 | 24.5 | 30.1 | 54.7 | 64.5 | 49.0 | 34.2 | 53.2 | 56.5 |
| PL | ✗ | 49.9 | 53.4 | 56.7 | 46.7 | 46.1 | 56.6 | 51.4 | 52.6 | 60.2 | 68.1 | 77.7 | 64.5 | 51.0 | 69.0 | 68.8 |
| FOA | ✗ | 46.5 | 47.0 | 51.1 | 44.8 | 45.5 | 52.3 | 48.1 | 49.7 | 57.9 | 68.0 | 76.4 | 63.6 | 51.7 | 62.2 | 65.3 |
| Tent | ✗ | 52.4 | 56.2 | 58.7 | 50.9 | 51.1 | 57.7 | 52.7 | 54.7 | 60.5 | 68.4 | 77.3 | 64.6 | 53.8 | 69.3 | 68.6 |
|  | ✓ | 54.7 | 59.0 | 60.3 | 51.2 | 53.7 | 60.6 | 57.4 | 64.1 | 65.8 | 71.0 | 78.6 | 66.9 | 62.7 | 71.8 | 70.5 |
|  | Improve | △2.3 | △2.8 | △1.6 | △0.4 | △2.6 | △2.9 | △4.7 | △9.5 | △5.3 | △2.6 | △1.3 | △2.3 | △9.0 | △2.5 | △1.9 |
| EATA | ✗ | 55.9 | 59.5 | 60.9 | 56.2 | 59.2 | 63.0 | 61.6 | 65.7 | 67.5 | 72.5 | 78.6 | 66.8 | 67.1 | 72.2 | 71.7 |
|  | ✓ | 57.9 | 60.6 | 61.5 | 57.0 | 59.8 | 63.7 | 62.3 | 67.2 | 68.3 | 73.7 | 78.8 | 69.7 | 68.6 | 73.1 | 71.8 |
|  | Improve | △2.0 | △1.0 | △0.6 | △0.8 | △0.6 | △0.8 | △0.7 | △1.5 | △0.8 | △1.2 | △0.3 | △2.9 | △1.4 | △0.9 | △0.1 |
| SAR | ✗ | 52.0 | 54.8 | 56.2 | 50.2 | 52.3 | 56.1 | 52.8 | 50.8 | 26.5 | 0.1 | 3.1 | 0.1 | 0.1 | 0.1 | 0.3 |
|  | ✓ | 56.0 | 60.1 | 61.1 | 56.4 | 58.1 | 62.9 | 60.4 | 66.1 | 67.4 | 72.2 | 78.7 | 68.0 | 66.0 | 72.6 | 70.9 |
|  | Improve | △4.0 | △5.3 | △4.9 | △6.2 | △5.8 | △6.8 | △7.6 | △15.3 | △41.0 | △72.1 | △75.6 | △67.9 | △65.9 | △72.5 | △70.6 |
| CoTTA | ✗ | 40.3 | 49.2 | 57.1 | 39.8 | 50.4 | 55.6 | 48.3 | 53.1 | 61.3 | 63.9 | 73.3 | 62.0 | 56.5 | 67.5 | 66.7 |
|  | ✓ | 49.7 | 61.7 | 64.2 | 45.7 | 57.0 | 59.0 | 51.1 | 58.2 | 63.1 | 66.0 | 73.4 | 62.9 | 58.0 | 68.9 | 68.2 |
|  | Improve | △9.4 | △12.4 | △7.1 | △5.9 | △6.6 | △3.4 | △2.8 | △5.1 | △1.8 | △2.1 | △0.1 | △1.0 | △1.5 | △1.3 | △1.5 |

**Ablation study (Q4).** We conduct the ablation study on different components in our COME, i.e., with and without the uncertainty constraint in Eq. 6. The experimental results are shown in Table 5, where SL indicates minimizing entropy of the subjective logic opinion and UC means the uncertainty constraint described by Eq. 6. Compared with non-constrained optimization, naively minimizing the entropy of subjective opinion can only slightly improve uncertainty estimation performance. Combining with the uncertainty constraint, the average and worst-case accuracy can be both substantially improved, which indicates the optimal design of our COME.

**Influence of different hyper-parameters (Q5).** We conduct additional experiments to investigate the influence of different hyperparameters. The results are in Table 12 and Table 13. Our COME generally outperforms EM with moderate hyperparamters. For $\tau$, as we mentioned before, it actually controls the magnitude of uncertainty mass. If we have some prior knowledge that most test samples

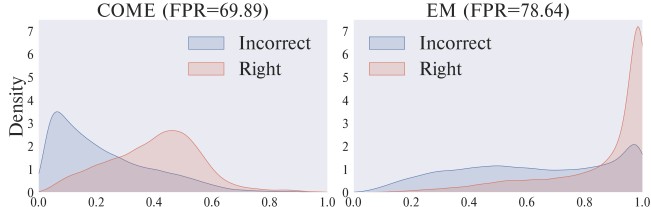

Figure 2: Distribution of model confidence.

Table 5: Ablation study.

| SL | UC | Acc↑ Mean | Acc↑ Worst | FPR↓ Mean | FPR↓ Worst |
|---|---|---|---|---|---|
| ✗ | ✗ | 52.8 | 10.6 | 70.2 | 95.3 |
| ✓ | ✗ | 52.7 | 10.4 | 70.0 | 94.9 |
| ✗ | ✓ | 60.9 | 25.5 | 68.0 | 93.0 |
| ✓ | ✓ | 61.2 | 51.7 | 67.3 | 71.4 |

should be rejected to adapt during TTA, we should choose a relatively small $\tau$ and making the model confidence more conservative in this circumstance. As for $\delta$ (or $p$) which represents the tolerance of uncertainty divergence, it should be selected per need by the user via trial and error: if users are extremely cautious about unreliable TTA, $\delta$ should be tuned down and p should be tuned up; otherwise, if a better performance is required. Besides, we introduce an additional hyperparameter $\lambda$ to the learning objective. The experiments results in Table 12 shows that an additional performance improvement can be observed when $\lambda \in [1, 100]$.

Table 6: Additional results with different $\lambda$. We report the accuracy when our COME is equipped to Tent on ImageNet-C Gaussian noise level 5 with different $\lambda$. The accuracy of the original Tent using entropy minimization is 52.6.

| $\lambda$ | 0.10 | 1.00 | 10.0 | 30.0 | 50.0 | 80.0 | 100 | 150 |
|---|---|---|---|---|---|---|---|---|
| Acc. | $53.7_{\pm0.1}$ | $53.9_{\pm0.0}$ | $54.6_{\pm0.1}$ | $55.3_{\pm0.1}$ | $55.8_{\pm0.1}$ | $56.2_{\pm0.1}$ | $56.1_{\pm0.1}$ | $10.6_{\pm0.3}$ |

Table 7: Additional results with different hyperparameters. We report the accuracy on ImageNet-C Gaussian noise level 5 with different hyperparameters. We implement our COME with Tent. The accuracy of the original Tent using entropy minimization is 52.6.

| | $p = 1$ | $p = 2$ | $p = 3$ | $p = \infty$ |
|---|---|---|---|---|
| $\tau = 0.5$ | $37.8_{\pm0.0}$ | $38.5_{\pm0.0}$ | $39.1_{\pm0.0}$ | $41.1_{\pm0.0}$ |
| $\tau = 1.0$ | $53.5_{\pm0.0}$ | $53.8_{\pm0.1}$ | $53.2_{\pm0.0}$ | $47.2_{\pm0.1}$ |
| $\tau = 1.2$ | $54.7_{\pm0.0}$ | $54.7_{\pm0.1}$ | $54.0_{\pm0.1}$ | $48.9_{\pm0.0}$ |
| $\tau = 1.5$ | $54.2_{\pm0.2}$ | $54.2_{\pm0.1}$ | $53.2_{\pm0.1}$ | $49.1_{\pm0.1}$ |

# 6 CONCLUSION

In this paper, we propose a novel learning principle called COME to improve existing TTA methods. COME explicitly models the uncertainty raising upon unreliable test samples using the theory of evidence, and then regularizes the model to favor of conservative prediction confidence during inference time. Our method takes inspiration from Bayesian framework, and consistently outperforms previous EM-based TTA methods on commonly-used benchmarks. The simplicity of the uncertainty regularization used in our implementation is both an advantage and a limitation. On the one hand, constraining the uncertainty mass close to the pretrain model is easy-to-deployed and meets the efficiency requirement of TTA. On the other hand, this regularization may be less effective when the pretrained model is also overconfident. We identify this as a limitation of our work. Exploring more effective regularization techniques for better trade-off between the practical requirements of TTA and accurate uncertainty estimation can be a promising future direction. Besides, vision foundation models such as DINOv2 are more promising to fill domain gaps. The representation learned by these foundation models are naturally expected to be much more robust than ImageNet pretrained ResNet and ViT models we used in this manuscript. Investigating TTA techniques for more advanced vision foundation models are also a promising research direction.

## 7 ACKNOWLEDGMENT

This work was supported by the National Natural Science Foundation of China (62376193 and 61925602). The authors thank ICLR anonymous peer reviewers for their helpful suggestions.

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

APPENDICES

## A  PROOFS

To proof Lemma 1 and Theorem 1, we need the following lemma firstly.

**Lemma 2.** *Let $p, q$ be two real numbers. Assuming that $p \leq q$, then the p-norm (also called $\ell^p$-norm) and q-norm of vector $x = (x_1, \cdots, x_n)$ satisfied*

$$||x||_p \leq n^{(1/q-1/p)}||x||_q. \tag{10}$$

*where $n$ is the length of the vector.*

*Proof.* Recall Hölder's inequality

$$\sum_{i=1}^{n} |a_i||b_i| \leq (\sum_{i=1}^{n} |a_i|^r)^{1/r}(\sum_{i=1}^{n} |b_i|^{\frac{r}{r-1}})^{1-\frac{1}{r}}. \tag{11}$$

Apply this inequality to the case that $|a_i| = |x_i|^p$, $|b_i| = 1$ and $r = q/p \geq 1$, we can derive to

$$\sum_{i=1}^{n} |a_i||b_i| \leq (\sum_{i=1}^{n} ((x_i)^p)^{\frac{q}{p}})^{\frac{p}{q}}(\sum_{i=1}^{n} 1^{\frac{q}{q-p}})^{1-\frac{p}{q}} = (\sum_{i=1}^{n} |x_i|^q)^{\frac{p}{q}} n^{1-\frac{p}{q}}. \tag{12}$$

Then we have

$$
\begin{aligned}
||x||_p &= (\sum_{i=1}^{n} |x_i|_p)^{1/p} \\
&\leq ((\sum_{i=1}^{n} |x_i|^q)^{\frac{p}{q}} n^{1-\frac{p}{q}})^{1/p} \\
&= (\sum_{i=1}^{n} |x_i|^q)^{\frac{1}{q}} n^{\frac{1}{p}-\frac{1}{q}} \\
&= n^{1/p-1/q} ||x||_q
\end{aligned}
\tag{13}
$$

$\square$

Now we proceed to proof our main results.

*Proof.* Proof of Lemma 1. Let $f_{\max} = \max_i f_i(x)$, then we have

$$
\exp(f_{\max}) \leq \sum_{i=1}^{K} \exp(f_i) \leq K \exp(f_{\max}),
\tag{14}
$$

Applying the logarithm to the inequality, then

$$
f_{\max} \leq \mathrm{LSE}(f) \leq f_{\max} + \log K,
\tag{15}
$$

where LSE is the shorthand of LogSumExp function, i.e., $\mathrm{LSE}(x) := \log \sum_{i=1}^{K} \exp x_i$.

Since we assume that all the elements in logits $x$ are all positive, then $f_{\max} = ||f||_\infty$. Thus combining with Lemma 1 we can derive that

$$
K^{-1/p} ||f||_p \leq \mathrm{LSE}(x) \leq ||f||_p + \log K,
\tag{16}
$$

Noted that $u = K/\mathrm{LSE}(f)$, then we have

$$
\frac{K}{||f||_p + \log K} \leq u \leq \frac{K^{1+1/p}}{||f||_p}.
\tag{17}
$$

$\square$

*Proof.* Proof of Theorem 1. Assuming that the uncertainty mass $u$ is constrained as

$$
u_0 - \delta \leq u \leq u_0 + \delta,
\tag{18}
$$

then the LSE function of model output is also bounded by

$$
\frac{K}{u_0 + \delta} \leq \mathrm{LSE}(f(x)) \leq \frac{K}{u_0 - \delta}.
\tag{19}
$$

Noted that

$$
\max f(x) \leq \mathrm{LSE}(f(x)),
\tag{20}
$$

and thus

$$
\max f(x) \leq \frac{K}{u_0 - \delta}.
\tag{21}
$$

According to Eq. 5, the model confidence is calculated by

$$
\max_k \mu_k(x) = \frac{\alpha_k}{S} = \frac{\exp f_{\max}}{\sum_{i=1}^{K} \exp f_i},
\tag{22}
$$

where $\alpha_i = \exp f_i(x)$.

Assuming the $f_j$ is the largest element in $f(x)$, then

$$\begin{aligned}
\max_k \mu_k(x) &= \frac{1}{1 + \sum_{i=1, i \neq j}^{K} \exp\left(f_i - f_{\max}\right)} \\
&\leq \frac{1}{1 + (K-1) \exp\left(f_{\min} - f_{\max}\right)} \\
&\leq \frac{1}{1 + (K-1) \exp\left(-\frac{K}{u_0 - \delta}\right)}
\end{aligned} \tag{23}$$

$\square$

## B    EXPERIMENTAL DETAILS

### B.1    DATASETS

**Covariate-shifted OOD generalization datasets.**    We conduct experiments on ImageNet-C (Hendrycks & Dietterich, 2019), which consists of 15 types of algorithmically generated corruptions from noise, blur, weather, and digital categories. Each type of corruption has 5 levels of severity, resulting in 75 distinct corruptions. Besides, we conduct additional experiments on ImageNet-R and ImageNet-S.

**Abnormal outliers for open-world TTA experiments.**    We follow the settings of (Zhang et al., 2023), where OpenImage-O (Wang et al., 2022a), SSB-hard (Vaze et al., 2022), Textures (Cimpoi et al., 2014), iNaturalist (Van Horn et al., 2018) and NINCO (Bitterwolf et al., 2023) are selected as outliers for ImageNet. ∘ OpenImage-O contains 17632 manually filtered images and is 7.8 × larger than the ImageNet dataset. ∘ SSB-hard is selected from ImageNet-21K. It consists of 49K images and 980 categories. ∘ Textures (Describable Textures Dataset, DTD) consists of 5,640 images depicting natural textures. ∘ iNaturalist consists of 859000 images from over 5000 different species of plants and animals. ∘ NINCO consists with a total of 5879 samples of 64 classes which are non-overlapped with ImageNet-C. These datasets are also used in previous open-set OOD generalization works (Gao et al., 2024; Bai et al., 2023; Liang et al.; Lee et al., 2023).

### B.2    IMPLEMENTATION DETAILS

**Pretrained models.**    The pretrained ViT model is ViT-Base (Dosovitskiy et al., 2021). The model is trained on the source ImageNet-1K training set and the model weights[3] are directly obtained from timm respository. Specifically, the pretrained ResNet model in  C.8 is ResNet-50-BN (He et al., 2016). The model is trained on the source ImageNet-1K training set and the model weights[4] are directly obtained from torchvision library.

**TEA[5] (Yige et al., 2024)** We follow all hyperparameters that are set in TEA unless it does not provide. Specifically, we use SGD as the update rule, with a momentum of 0.9, batch size of 64 and learning rate of 0.001/0.00025 for ViT/ResNet models. The trainable parameters are all affine parameters of layer/batch normalization layers for ViT/ResNet models.

**T3A[6] (Iwasawa & Matsuo, 2021)** We follow all hyperparameters that are set in T3A unless it does not provide. Specifically, the batch size is set to 64. The number of supports to restore M is set to 20 for all experiments.

**LAME[7] (Boudiaf et al., 2022)** We follow all hyperparameters that are set in LAME unless it does not provide. For fair comparison, we maintain a consistent batch size of 64 for LAME, aligning it with the same batch size used by other methods in our evaluation. We use the kNN affinity matrix with the value of $k = 5$.

---

[3]https://huggingface.co/google/vit-base-patch16-224

[4]https://download.pytorch.org/models/resnet50-19c8e357.pth

[5]https://github.com/yuanyige/tea

[6]https://github.com/matsuolab/T3A

[7]https://github.com/fiveai/LAME

**FOA**[8] **(Niu et al., 2024)** We follow all hyperparameters that are set in FOA unless it does not provide. Specifically, the batch size is set to 64. The number of supports to restore M is set to 20 for all experiments.

**Tent**[9] **(Wang et al., 2021)** We follow all hyperparameters that are set in Tent unless it does not provide. Specifically, we use SGD as the update rule, with a momentum of 0.9, batch size of 64 and learning rate of 0.001/0.00025 for ViT/ResNet models. The trainable parameters are all affine parameters of layer/batch normalization layers for ViT/ResNet models.

**EATA**[10] **(Niu et al., 2022)** We follow all hyperparameters that are set in EATA. Specifically, the entropy constant $E_0$ (for reliable sample identification) is set to $0.4 \times ln1000$, where 1000 is the number of task classes. The $\epsilon$ for redundant sample identification is set to 0.05. The trade-off parameter $\beta$ for entropy loss and regularization loss is set to 2,000. The number of pre-collected in-distribution test samples for Fisher importance calculation is 2,000. We use SGD as the update rule, with a momentum of 0.9, batch size of 64 and learning rate of 0.001/0.00025 for ViT/ResNet models. The trainable parameters are all affine parameters of layer/batch normalization layers for ViT/ResNet models.

**SAR**[11] **(Niu et al., 2023)** We follow all hyperparameters that are set in SAR. Specifically, the entropy threshold $E_0$ is set to $0.4 \times ln1000$, where 1000 is the number of task classes. We use SGD as the update rule, with a momentum of 0.9, batch size of 64 and learning rate of 0.001/0.00025 for ViT/ResNet models. For model recovery, we follow all strategy that are set in SAR(except for the experiments of life-long).The trainable parameters are all affine parameters of layer/batch normalization layers for ViT/ResNet models.

**CoTTA**[12] **(Wang et al., 2022b)** We follow all hyperparameters that are set in CoTTA unless it does not provide. Specifically, we use SGD as the update rule, with a momentum of 0.9, batch size of 64 and learning rate of 0.001/0.01 for ViT/ResNet models. The augmentation threshold $p_{th}$ is set to 0.1. For images below threshold, we conduct 32 augmentations including color jitter, random affine, Gaussian blur, random horizonal flip, and Gaussian noise. The restoration probability of is set to 0.01 and the EMA factor $\alpha$ for teacher update is set to 0.999. The trainable parameters are all affine parameters of layer/batch normalization layers for ViT/ResNet models.

**MEMO**[13] **(Zhang et al., 2022)** We follow all hyperparameters that are set in MEMO. Specifically, we use the AugMix[14] (Hendrycks et al., 2020) as a set of data augmentations and the augmentation size is set to 32. We use SGD as the optimizer,with learning rate 0.00025 and no weight decay. The trainable parameters are the entire model.

**Source of standard deviation.** For all the experiments, we run multiple times and report the average performance and standard deviation. The source of the standard deviation consists 1) the order in which the test mini-batches coming online and 2) the randomness of the stochastic optimization methods, e.g., SGD, Adam. Since in TTA setting, the model is initialized from the publicly available pretrained model weights (i.e., via-base-patch16-224 from timm and resnet50 from PyTorch), there is no randomness introduced by model initialization.

## C  ADDITIONAL RESULTS

### C.1  FULL RESULTS OF ACCURACY WITH STANDARD DEVIATION (SUPPLEMENTARY TO TABLE 1)

We provide the full results with standard deviation as supplementary to Table 1 in Table 8. The results demonstrate that our COME method consistently achieves better performance than its counterparts.

---

[8]https://github.com/mr-eggplant/FOA

[9]https://github.com/DequanWang/tent

[10]https://github.com/mr-eggplant/EATA

[11]https://github.com/mr-eggplant/SAR

[12]https://github.com/qinenergy/cotta

[13]https://github.com/zhangmarvin/memo

[14]https://github.com/google-research/augmix

Table 8: Classification accuracy comparison with standard deviation on ImageNet-C (level 5). Substantial ($\geq 0.5$) improvement and degradation compared to the baseline are highlighted in blue or brown respectively.

| Methods | COME | Noise | | | Blur | | | | Weather | | | | Digital | | | |
|---|---|---|---|---|---|---|---|---|---|---|---|---|---|---|---|---|
| | | Gauss. | Shot | Impul. | Defoc | Glass | Motion | Zoom | Snow | Frost | Fog | Brit. | Contr. | Elast. | Pixel | JPEG |
| No Adapt | ✗ | $35.1_{\pm0.0}$ | $32.2_{\pm0.0}$ | $35.9_{\pm0.0}$ | $31.4_{\pm0.0}$ | $25.3_{\pm0.0}$ | $39.4_{\pm0.0}$ | $31.6_{\pm0.0}$ | $24.5_{\pm0.0}$ | $30.1_{\pm0.0}$ | $54.7_{\pm0.0}$ | $64.5_{\pm0.0}$ | $49.0_{\pm0.0}$ | $34.2_{\pm0.0}$ | $53.2_{\pm0.0}$ | $56.5_{\pm0.0}$ |
| PL | ✗ | $49.9_{\pm0.2}$ | $48.8_{\pm0.2}$ | $50.9_{\pm0.1}$ | $48.5_{\pm0.4}$ | $41.2_{\pm0.3}$ | $52.7_{\pm0.2}$ | $42.7_{\pm0.5}$ | $24.6_{\pm1.9}$ | $42.5_{\pm1.9}$ | $63.6_{\pm0.5}$ | $73.1_{\pm0.1}$ | $65.3_{\pm0.2}$ | $44.1_{\pm0.6}$ | $63.9_{\pm0.1}$ | $62.6_{\pm0.1}$ |
| T3A | ✗ | $34.6_{\pm0.0}$ | $31.5_{\pm0.0}$ | $35.5_{\pm0.1}$ | $32.7_{\pm0.0}$ | $27.5_{\pm0.0}$ | $40.7_{\pm0.1}$ | $33.5_{\pm0.1}$ | $25.6_{\pm0.0}$ | $30.8_{\pm0.2}$ | $56.5_{\pm0.1}$ | $64.9_{\pm0.0}$ | $50.8_{\pm0.1}$ | $38.0_{\pm0.1}$ | $54.3_{\pm0.0}$ | $58.4_{\pm0.0}$ |
| TEA | ✗ | $44.5_{\pm0.1}$ | $39.3_{\pm0.1}$ | $45.8_{\pm0.2}$ | $37.6_{\pm0.4}$ | $35.4_{\pm0.7}$ | $46.4_{\pm0.2}$ | $31.4_{\pm9.7}$ | $8.9_{\pm0.3}$ | $46.4_{\pm0.3}$ | $60.1_{\pm0.2}$ | $72.5_{\pm0.1}$ | $59.5_{\pm0.8}$ | $45.7_{\pm0.3}$ | $62.3_{\pm0.1}$ | $58.9_{\pm0.2}$ |
| LAME | ✗ | $34.8_{\pm0.0}$ | $31.9_{\pm0.0}$ | $35.5_{\pm0.0}$ | $30.9_{\pm0.0}$ | $24.4_{\pm0.0}$ | $38.9_{\pm0.1}$ | $30.7_{\pm0.0}$ | $23.4_{\pm0.0}$ | $29.5_{\pm0.0}$ | $53.3_{\pm0.0}$ | $64.2_{\pm0.0}$ | $41.0_{\pm0.1}$ | $32.7_{\pm0.0}$ | $52.8_{\pm0.0}$ | $56.0_{\pm0.0}$ |
| FOA | ✗ | $46.6_{\pm0.7}$ | $43.9_{\pm0.9}$ | $48.3_{\pm0.1}$ | $47.1_{\pm0.4}$ | $40.7_{\pm0.6}$ | $49.3_{\pm0.3}$ | $43.7_{\pm1.0}$ | $53.8_{\pm0.5}$ | $52.8_{\pm0.3}$ | $64.2_{\pm1.2}$ | $76.2_{\pm0.4}$ | $63.8_{\pm0.4}$ | $48.5_{\pm0.6}$ | $62.1_{\pm0.7}$ | $64.0_{\pm0.0}$ |
| Tent | ✗ | $52.5_{\pm0.1}$ | $52.1_{\pm0.1}$ | $53.4_{\pm0.1}$ | $52.8_{\pm0.1}$ | $47.4_{\pm0.5}$ | $56.7_{\pm0.0}$ | $47.4_{\pm0.1}$ | $10.5_{\pm1.2}$ | $26.4_{\pm2.1}$ | $67.2_{\pm0.1}$ | $74.3_{\pm0.1}$ | $67.3_{\pm0.0}$ | $50.4_{\pm0.3}$ | $66.5_{\pm0.1}$ | $64.6_{\pm0.0}$ |
| | ✓ | $53.9_{\pm0.1}$ | $53.9_{\pm0.0}$ | $55.2_{\pm0.1}$ | $55.8_{\pm0.1}$ | $51.8_{\pm0.1}$ | $59.8_{\pm0.0}$ | $52.6_{\pm0.0}$ | $58.4_{\pm0.5}$ | $61.3_{\pm0.1}$ | $71.3_{\pm0.1}$ | $78.2_{\pm0.0}$ | $68.8_{\pm0.1}$ | $57.9_{\pm0.4}$ | $70.5_{\pm0.1}$ | $68.2_{\pm0.1}$ |
| | Improve | $\triangle1.4$ | $\triangle1.9$ | $\triangle1.8$ | $\triangle2.9$ | $\triangle4.4$ | $\triangle3.2$ | $\triangle5.2$ | $\triangle47.9$ | $\triangle34.9$ | $\triangle4.1$ | $\triangle3.9$ | $\triangle1.5$ | $\triangle7.6$ | $\triangle4.0$ | $\triangle3.6$ |
| EATA | ✗ | $56.0_{\pm0.2}$ | $56.1_{\pm0.3}$ | $57.1_{\pm0.1}$ | $54.5_{\pm1.9}$ | $54.8_{\pm1.7}$ | $59.6_{\pm1.6}$ | $58.7_{\pm0.1}$ | $61.8_{\pm0.3}$ | $60.1_{\pm0.1}$ | $71.4_{\pm0.2}$ | $75.3_{\pm0.0}$ | $68.5_{\pm0.1}$ | $62.7_{\pm0.3}$ | $69.0_{\pm0.3}$ | $66.5_{\pm0.2}$ |
| | ✓ | $56.1_{\pm0.1}$ | $56.5_{\pm0.1}$ | $57.2_{\pm0.0}$ | $58.0_{\pm0.1}$ | $57.9_{\pm0.2}$ | $62.6_{\pm0.1}$ | $59.3_{\pm0.2}$ | $65.6_{\pm0.2}$ | $63.5_{\pm0.3}$ | $72.6_{\pm0.1}$ | $78.0_{\pm0.1}$ | $69.5_{\pm0.1}$ | $66.6_{\pm0.3}$ | $72.5_{\pm0.2}$ | $70.5_{\pm0.1}$ |
| | Improve | $\triangle0.1$ | $\triangle0.3$ | $\triangle0.1$ | $\triangle3.4$ | $\triangle3.1$ | $\triangle3.1$ | $\triangle0.6$ | $\triangle3.9$ | $\triangle3.4$ | $\triangle1.3$ | $\triangle2.7$ | $\triangle1.0$ | $\triangle3.9$ | $\triangle3.5$ | $\triangle4.0$ |
| SAR | ✗ | $51.9_{\pm0.1}$ | $51.7_{\pm0.1}$ | $52.8_{\pm0.1}$ | $51.5_{\pm0.5}$ | $48.9_{\pm0.2}$ | $55.5_{\pm0.1}$ | $49.5_{\pm0.2}$ | $22.2_{\pm0.8}$ | $46.9_{\pm2.2}$ | $66.2_{\pm0.4}$ | $72.9_{\pm0.1}$ | $65.8_{\pm0.1}$ | $50.9_{\pm0.5}$ | $64.0_{\pm0.0}$ | $62.8_{\pm0.2}$ |
| | ✓ | $56.4_{\pm0.1}$ | $56.6_{\pm0.1}$ | $57.4_{\pm0.1}$ | $58.3_{\pm0.2}$ | $56.9_{\pm0.1}$ | $62.9_{\pm0.1}$ | $58.3_{\pm0.2}$ | $65.3_{\pm0.1}$ | $64.5_{\pm0.1}$ | $72.7_{\pm0.1}$ | $78.5_{\pm0.0}$ | $69.6_{\pm0.0}$ | $64.0_{\pm0.2}$ | $71.9_{\pm0.1}$ | $69.7_{\pm0.1}$ |
| | Improve | $\triangle4.5$ | $\triangle4.9$ | $\triangle4.6$ | $\triangle6.8$ | $\triangle8.1$ | $\triangle7.4$ | $\triangle8.8$ | $\triangle43.1$ | $\triangle17.5$ | $\triangle6.5$ | $\triangle5.5$ | $\triangle3.8$ | $\triangle13.2$ | $\triangle7.9$ | $\triangle6.9$ |
| COTTA | ✗ | $40.3_{\pm0.2}$ | $37.6_{\pm0.1}$ | $41.7_{\pm0.1}$ | $34.3_{\pm0.4}$ | $28.3_{\pm0.7}$ | $44.0_{\pm0.1}$ | $35.6_{\pm0.2}$ | $38.0_{\pm0.1}$ | $43.0_{\pm0.2}$ | $58.8_{\pm0.3}$ | $70.3_{\pm0.3}$ | $58.4_{\pm0.5}$ | $39.8_{\pm0.2}$ | $58.1_{\pm0.2}$ | $59.9_{\pm0.1}$ |
| | ✓ | $43.5_{\pm0.0}$ | $40.9_{\pm0.3}$ | $45.5_{\pm0.4}$ | $36.9_{\pm0.2}$ | $29.7_{\pm0.2}$ | $48.1_{\pm1.2}$ | $37.8_{\pm0.3}$ | $40.7_{\pm1.4}$ | $42.0_{\pm0.4}$ | $62.3_{\pm0.6}$ | $73.6_{\pm0.2}$ | $58.9_{\pm2.4}$ | $42.8_{\pm0.3}$ | $63.5_{\pm0.2}$ | $63.8_{\pm0.1}$ |
| | Improve | $\triangle3.1$ | $\triangle3.4$ | $\triangle3.8$ | $\triangle2.6$ | $\triangle1.4$ | $\triangle4.0$ | $\triangle2.2$ | $\triangle2.7$ | $\triangledown1.0$ | $\triangle3.5$ | $\triangle3.3$ | $\triangle0.5$ | $\triangle3.0$ | $\triangle5.4$ | $\triangle3.9$ |

## C.2 Full results of open-world TTA (supplementary to Table 3)

We provide the full results with standard deviation as supplementary to Table 3 in Tableï¡đ9. The results demonstrate that our COME method consistently achieves better performance than its counterparts.

Table 9: Classification and uncertainty estimation comparisons with standard deviation under **open-world** TTA settings, where $P^{\text{test}}$ is a mixture of both covariate-shifted samples (Gaussian noise of severity level 3).

| Method | COME | None | | NINCO | | iNaturist | | SSB-Hard | | Texture | | Places | |
|---|---|---|---|---|---|---|---|---|---|---|---|---|---|
| | | Acc↑ | FPR↓ | Acc↑ | FPR↓ | Acc↑ | FPR↓ | Acc↑ | FPR↓ | Acc↑ | FPR↓ | Acc↑ | FPR↓ |
| No Adapt | ✗ | $64.4_{\pm0.0}$ | $63.7_{\pm0.0}$ | $64.6_{\pm0.3}$ | $70.3_{\pm1.0}$ | $64.4_{\pm0.0}$ | $69.5_{\pm0.0}$ | $64.4_{\pm0.0}$ | $72.5_{\pm0.1}$ | $64.7_{\pm0.2}$ | $65.0_{\pm0.4}$ | $64.5_{\pm0.3}$ | $57.4_{\pm0.6}$ |
| PL | ✗ | $69.1_{\pm0.1}$ | $63.0_{\pm0.3}$ | $65.6_{\pm0.0}$ | $70.9_{\pm1.7}$ | $68.8_{\pm0.1}$ | $69.3_{\pm0.2}$ | $68.3_{\pm0.0}$ | $76.2_{\pm0.3}$ | $66.1_{\pm0.1}$ | $65.7_{\pm0.3}$ | $67.1_{\pm0.3}$ | $59.1_{\pm0.6}$ |
| T3A | ✗ | $64.4_{\pm0.0}$ | $71.0_{\pm0.3}$ | $64.3_{\pm0.2}$ | $71.4_{\pm1.6}$ | $64.1_{\pm0.1}$ | $74.1_{\pm0.6}$ | $63.7_{\pm0.0}$ | $80.1_{\pm0.4}$ | $64.4_{\pm0.2}$ | $67.4_{\pm1.4}$ | $64.0_{\pm0.3}$ | $69.6_{\pm0.1}$ |
| TEA | ✗ | $64.1_{\pm0.3}$ | $63.3_{\pm0.5}$ | $60.2_{\pm0.6}$ | $72.9_{\pm0.3}$ | $62.4_{\pm0.1}$ | $74.5_{\pm0.2}$ | $63.5_{\pm0.1}$ | $79.3_{\pm0.2}$ | $60.6_{\pm0.4}$ | $69.0_{\pm0.9}$ | $62.0_{\pm0.2}$ | $65.8_{\pm1.0}$ |
| LAME | ✗ | $64.1_{\pm0.0}$ | $64.1_{\pm0.3}$ | $64.2_{\pm0.3}$ | $72.3_{\pm0.1}$ | $64.1_{\pm0.0}$ | $72.2_{\pm0.2}$ | $64.1_{\pm0.0}$ | $73.9_{\pm0.1}$ | $64.6_{\pm0.2}$ | $68.5_{\pm0.3}$ | $64.3_{\pm0.3}$ | $61.4_{\pm0.3}$ |
| FOA | ✗ | $67.8_{\pm0.0}$ | $62.0_{\pm0.6}$ | $65.9_{\pm0.3}$ | $69.2_{\pm0.9}$ | $67.6_{\pm0.2}$ | $66.8_{\pm0.5}$ | $67.5_{\pm0.2}$ | $76.0_{\pm0.3}$ | $65.8_{\pm0.0}$ | $60.6_{\pm0.5}$ | $66.7_{\pm0.1}$ | $52.8_{\pm0.9}$ |
| Tent | ✗ | $70.9_{\pm0.1}$ | $63.6_{\pm0.5}$ | $66.4_{\pm0.3}$ | $71.5_{\pm0.3}$ | $70.0_{\pm0.1}$ | $70.8_{\pm0.4}$ | $69.8_{\pm0.1}$ | $77.4_{\pm0.1}$ | $66.7_{\pm0.4}$ | $67.3_{\pm0.7}$ | $68.0_{\pm0.4}$ | $60.8_{\pm0.8}$ |
| | ✓ | $72.6_{\pm0.0}$ | $64.9_{\pm0.1}$ | $69.0_{\pm0.1}$ | $63.9_{\pm0.3}$ | $72.6_{\pm0.1}$ | $63.6_{\pm0.1}$ | $72.7_{\pm0.0}$ | $70.6_{\pm0.0}$ | $68.5_{\pm0.0}$ | $60.5_{\pm0.0}$ | $70.7_{\pm0.0}$ | $46.3_{\pm0.2}$ |
| | Improve | $\triangle1.7$ | $\triangle1.3$ | $\triangle2.6$ | $\triangledown7.6$ | $\triangle2.6$ | $\triangledown7.2$ | $\triangle2.8$ | $\triangledown6.8$ | $\triangle1.8$ | $\triangledown6.8$ | $\triangle2.6$ | $\triangledown14.5$ |
| EATA | ✗ | $70.2_{\pm0.1}$ | $63.5_{\pm0.1}$ | $66.2_{\pm0.1}$ | $68.8_{\pm0.2}$ | $70.2_{\pm0.1}$ | $71.5_{\pm0.1}$ | $69.9_{\pm0.1}$ | $77.5_{\pm0.1}$ | $67.6_{\pm0.5}$ | $66.5_{\pm0.5}$ | $68.6_{\pm0.2}$ | $62.0_{\pm0.2}$ |
| | ✓ | $73.3_{\pm0.0}$ | $63.1_{\pm0.3}$ | $70.1_{\pm0.0}$ | $60.5_{\pm0.0}$ | $73.2_{\pm0.0}$ | $63.2_{\pm0.1}$ | $73.0_{\pm0.0}$ | $70.6_{\pm0.1}$ | $70.3_{\pm0.2}$ | $56.8_{\pm0.7}$ | $72.3_{\pm0.0}$ | $45.8_{\pm0.2}$ |
| | Improve | $\triangle3.1$ | $\triangledown0.4$ | $\triangle3.9$ | $\triangledown8.2$ | $\triangle3.0$ | $\triangledown8.3$ | $\triangle3.1$ | $\triangledown6.9$ | $\triangle2.7$ | $\triangledown9.7$ | $\triangle3.7$ | $\triangledown16.2$ |
| SAR | ✗ | $69.6_{\pm0.1}$ | $62.3_{\pm0.1}$ | $65.5_{\pm0.5}$ | $70.3_{\pm0.9}$ | $67.1_{\pm0.1}$ | $70.7_{\pm0.2}$ | $67.6_{\pm0.1}$ | $77.9_{\pm0.2}$ | $65.5_{\pm0.8}$ | $65.8_{\pm1.0}$ | $66.2_{\pm0.1}$ | $59.8_{\pm1.0}$ |
| | ✓ | $73.1_{\pm0.0}$ | $62.3_{\pm0.4}$ | $70.0_{\pm0.2}$ | $65.9_{\pm0.2}$ | $73.3_{\pm0.1}$ | $65.3_{\pm0.0}$ | $73.5_{\pm0.0}$ | $71.5_{\pm0.3}$ | $69.4_{\pm0.0}$ | $59.3_{\pm0.0}$ | $72.0_{\pm0.2}$ | $48.5_{\pm0.9}$ |
| | Improve | $\triangle3.5$ | $\triangle0.1$ | $\triangle4.5$ | $\triangledown4.3$ | $\triangle6.2$ | $\triangledown5.4$ | $\triangle6.0$ | $\triangledown6.5$ | $\triangle3.9$ | $\triangledown6.5$ | $\triangle5.8$ | $\triangledown11.3$ |
| COTTA | ✗ | $67.7_{\pm0.1}$ | $63.4_{\pm0.1}$ | $65.3_{\pm0.4}$ | $70.4_{\pm0.8}$ | $70.4_{\pm0.2}$ | $69.5_{\pm0.1}$ | $70.1_{\pm0.1}$ | $75.8_{\pm0.4}$ | $65.7_{\pm0.2}$ | $65.0_{\pm0.9}$ | $67.0_{\pm0.3}$ | $58.8_{\pm0.3}$ |
| | ✓ | $70.4_{\pm0.1}$ | $62.8_{\pm0.2}$ | $66.5_{\pm0.3}$ | $68.0_{\pm0.6}$ | $72.4_{\pm0.0}$ | $72.9_{\pm0.5}$ | $72.1_{\pm0.1}$ | $78.5_{\pm0.2}$ | $66.4_{\pm0.2}$ | $63.8_{\pm0.7}$ | $68.9_{\pm0.4}$ | $54.5_{\pm0.5}$ |
| | Improve | $\triangle2.8$ | $\triangledown0.6$ | $\triangle1.2$ | $\triangledown2.4$ | $\triangle2.0$ | $\triangle3.5$ | $\triangle2.0$ | $\triangle2.7$ | $\triangle0.7$ | $\triangledown1.2$ | $\triangle1.9$ | $\triangledown4.4$ |

## C.3 Comparison with source-free domain adaption

There is a strong connection between Source-Free Domain Adaptation (SFDA) and Test-Time Adaptation (TTA). The primary difference is that TTA focuses on **online** adjusting during the testing. On the other hand, SFDA approaches generally perform **offline**. That is, the inference is deferred until the optimization is done. In contrast, our TTA method can achieve adaption and inference at the same time.

To further validate the applicability of our method, we report the classification accuracy on ImageNet-C Gaussian noise level 5 under **source-free domain adpation settings**. The results are in Table 10. The baselines we considered include pseudo label (PL), mutual information maximization (IM), and entropy minimization (EM) following (Liang et al., 2020). "-" means the model accuracy collapses to random guess level.

Table 10: Classification accuracy comparison on ImageNet-C Gaussian noise (level 5) under source-free domain adaption settings. "-" means the classification accuracy collapses to random guess level. The accuracy of the original pretrained model is 35.1.

| EPOCH COME | PL ✗ | EM ✗ | IM ✗ | TENT ✓ | EATA ✓ | SAR ✓ |
|---|---|---|---|---|---|---|
| 1 | $34.2_{\pm 3.2}$ | - | $60.7_{\pm 0.0}$ | $66.5_{\pm 0.0}$ | $66.4_{\pm 0.3}$ | $68.5_{\pm 0.1}$ |
| 2 | - | - | $63.8_{\pm 0.1}$ | $68.4_{\pm 0.0}$ | $67.2_{\pm 0.2}$ | $69.7_{\pm 0.1}$ |
| 3 | - | - | $65.1_{\pm 0.1}$ | $69.2_{\pm 0.0}$ | $67.8_{\pm 0.2}$ | $70.2_{\pm 0.1}$ |

## C.4 COMPARISON WITH OTHER OTHER BAYESIAN METHODS

There exists a few Beysian inspired TTA methods closely related to our method. (Zhou & Levine, 2021) explores Bayesian model ensembling Zhou & Levine (2021) for TTA, which introduces noticeable inference latency. Since (Zhou & Levine, 2021) has not made their source code publicly available. For a fair comparison, we implement the proposed COME using the same backbone (ResNet50-v2) and dataset (ImageNet-C) as in (Zhou & Levine, 2021) and report the average accuracy of all corruptions and levels. The results of (Zhou & Levine, 2021) are directly copied from the original paper. The results are in Table 11.

Table 11: Classification accuracy comparison with other bayesian insipired TTA methods on ImageNet-C under TTA settings averaged over all corruption types and levels. The result of BACS is copied from the original paper.

| COME | No Adapt ✗ | BN ✗ | BACS ✗ | TENT ✗ | TENT ✓ | EATA ✓ |
|---|---|---|---|---|---|---|
| | $47.3_{\pm 0.0}$ | $47.3_{\pm 0.0}$ | $56.1$ | $48.9_{\pm 0.0}$ | $51.1_{\pm 0.0}$ | $58.1_{\pm 0.0}$ |

## C.5 INFLUENCE OF DIFFERENT HYPERPARAMETERS

We conduct additional experiments to investigate the influence of different hyperparameters. The results are in Table 12 and Table 13. Our COME generally outperforms EM with moderate hyperparamters. For $\tau$, as we mentioned before, it actually controls the magnitude of uncertainty mass. If we have some prior knowledge that most test samples should be rejected to adapt during TTA, we should choose a relatively small $\tau$ and making the model confidence more conservative in this circumstance. As for $\delta$ (or $p$) which represents the tolerance of uncertainty divergence, it should be selected per need by the user via trial and error: if users are extremely cautious about unreliable TTA, $\delta$ should be tuned down and p should be tuned up; otherwise, if a better performance is required. Besides, we introduce an additional hyperparameter $\lambda$ to the learning objective. The experiments results in Table 12 shows that an additional performance improvement can be observed when $\lambda \in [1, 100]$.

Table 12: Additional results with different $\lambda$. We report the accuracy when our COME is equipped to Tent on ImageNet-C Gaussian noise level 5 with different $\lambda$. The accuracy of the original Tent using entropy minimization is 52.6.

| $\lambda$ | 0.10 | 1.00 | 10.0 | 30.0 | 50.0 | 80.0 | 100 | 150 |
|---|---|---|---|---|---|---|---|---|
| Acc. | $53.7_{\pm 0.1}$ | $53.9_{\pm 0.0}$ | $54.6_{\pm 0.1}$ | $55.3_{\pm 0.1}$ | $55.8_{\pm 0.1}$ | $56.2_{\pm 0.1}$ | $56.1_{\pm 0.1}$ | $10.6_{\pm 0.3}$ |

## C.6 COMPARISON ON IMAGENET-R AND IMAGENET-S

We introduce two more additional datasets, i.e., ImageNet-R and ImageNet-S, to further evaluate the proposed method on commonly used OOD generalization benchmarks. The results are in Table 14 and Table 15.

Table 13: Additional results with different hyperparameters. We report the accuracy on ImageNet-C Gaussian noise level 5 with different hyperparameters. We implement our COME with Tent. The accuracy of the original Tent using entropy minimization is 52.6.

|  | $p = 1$ | $p = 2$ | $p = 3$ | $p = \infty$ |
|---|---|---|---|---|
| $\tau = 0.5$ | $37.8_{\pm 0.0}$ | $38.5_{\pm 0.0}$ | $39.1_{\pm 0.0}$ | $41.1_{\pm 0.0}$ |
| $\tau = 1.0$ | $53.5_{\pm 0.0}$ | $53.8_{\pm 0.1}$ | $53.2_{\pm 0.0}$ | $47.2_{\pm 0.1}$ |
| $\tau = 1.2$ | $54.7_{\pm 0.0}$ | $54.7_{\pm 0.1}$ | $54.0_{\pm 0.1}$ | $48.9_{\pm 0.0}$ |
| $\tau = 1.5$ | $54.2_{\pm 0.2}$ | $54.2_{\pm 0.1}$ | $53.2_{\pm 0.1}$ | $49.1_{\pm 0.1}$ |

Table 14: Additional results on ImageNet-R

|  | Tent | EATA | SAR | COTTA |
|---|---|---|---|---|
| EM | $37.73 \pm 0.03$ | $36.11 \pm 0.12$ | $51.90 \pm 0.35$ | $36.04 \pm 0.06$ |
| COME | $39.05 \pm 0.05$ | $38.22 \pm 0.23$ | $56.40 \pm 0.18$ | $37.39 \pm 0.08$ |

### C.7 In-Distribution performance

We compare the in-distribution performance of proposed COME to EM-based methods. As shown in Table 16, our method consistently outperforms entropy minimization.

### C.8 Comparison on ResNet-50

As shown in previous work (Niu et al., 2023), the TTA performance can be influenced by different model architectures, especially the type of normalization layers, i.e., batch normalization, group normalization, layer normalization, and instance normalization. To further evaluate the proposed method, we conduct additional experiments on ResNet-50 with batch normalization layers under open-world TTA settings. The experimental results in Table 17 show that our COME still achieves superior performance compared to entropy minimization learning principle when equipped to Tent.

### C.9 Time-consuming comparison

We compare the time-cost of proposed COME to EM-based methods and Non-EM based methods in Table 18. We run all the experiments on one single NVIDIA 4090 GPU. Our COME does not introduce noticeably extra cost of computation.

### C.10 Mixed distributional shifts performance.

We evaluate the proposed COME in two additional settings introduced by (Niu et al., 2023). These scenarios include 1) online imbalanced label distribution shifts, where the test data are sorted by class, and 2) mixed domain shifts, where the test data stream includes several randomly mixed domains with different distribution shifts. As shown in Table 19 and Table 20, our COME consistently outperforms entropy minimization with an exception of a slightly suboptimal uncertainty estimation performance compared to CoTTA.

### C.11 More visualization results (supplementary to Figure 1)

We provide more visualization results on two representative TTA methods, i.e., the seminal Tent (Wang et al., 2021) and recent SOTA SAR (Niu et al., 2023). We observe that our COME enjoys a more stable TTA progress with less risk of model collapse and overconfidence across various types of corruption. We test on ImageNet-C level 5.

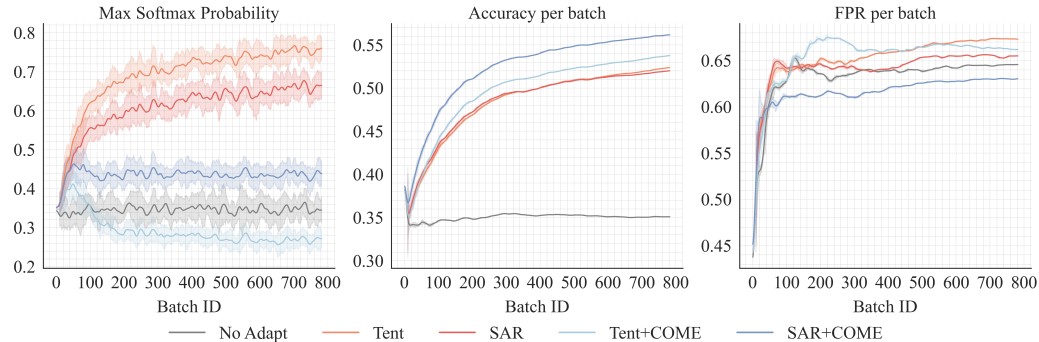

Figure 3: Comparison on two representative TTA methods on ImageNet-C under **Gaussian Noise** corruption of severity level 5. By contrast to EM, our COME establishes a stable TTA process with consistently improved classification accuracy and false positive rate.

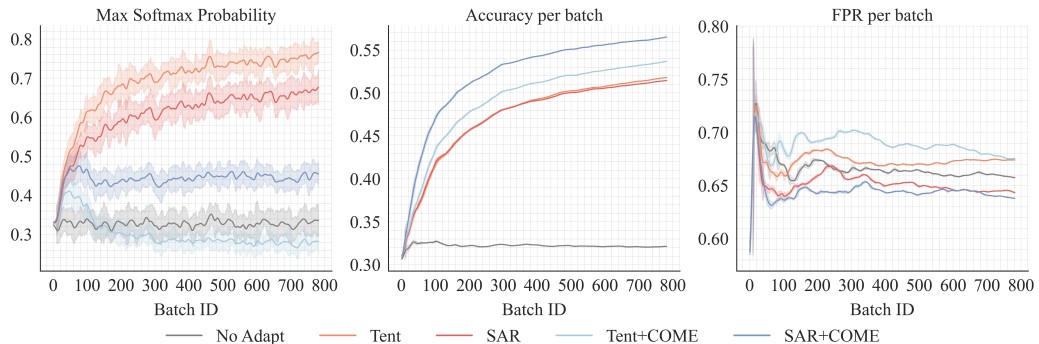

Figure 4: Comparison on two representative TTA methods on ImageNet-C under **Shot Noise** corruption of severity level 5. By contrast to EM, our COME establishes a stable TTA process with consistently improved classification accuracy and false positive rate.

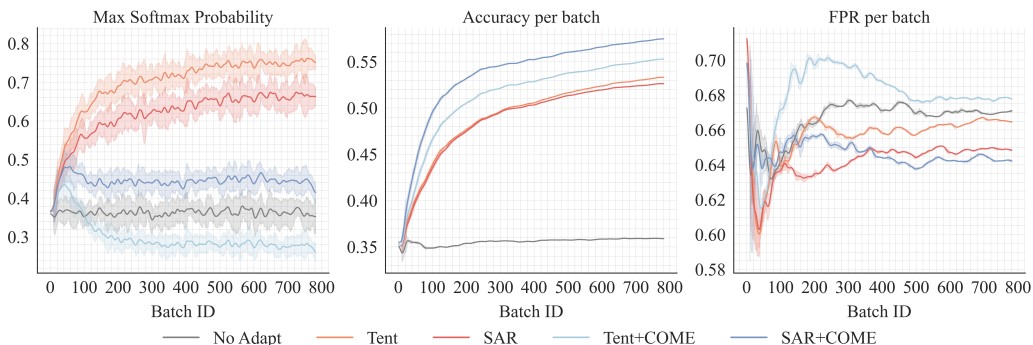

Figure 5: Comparison on two representative TTA methods on ImageNet-C under **Impulse Noise** corruption of severity level 5. By contrast to EM, our COME establishes a stable TTA process with consistently improved classification accuracy and false positive rate.

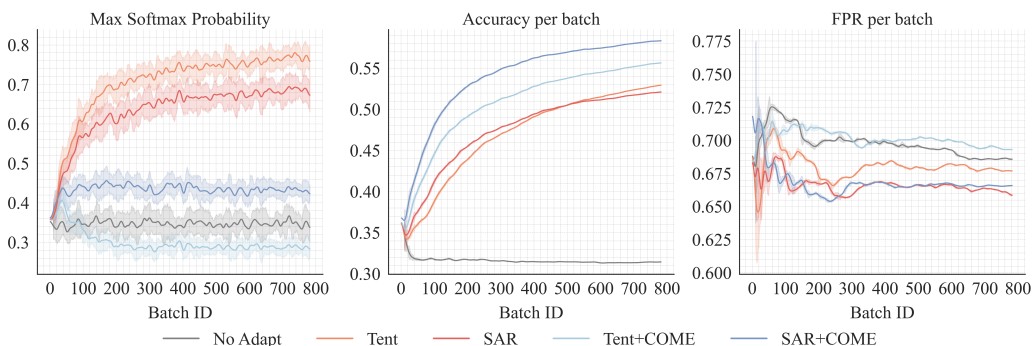

Figure 6: Comparison on two representative TTA methods on ImageNet-C under **Defocus Blur** corruption of severity level 5. By contrast to EM, our COME establishes a stable TTA process with consistently improved classification accuracy and false positive rate.

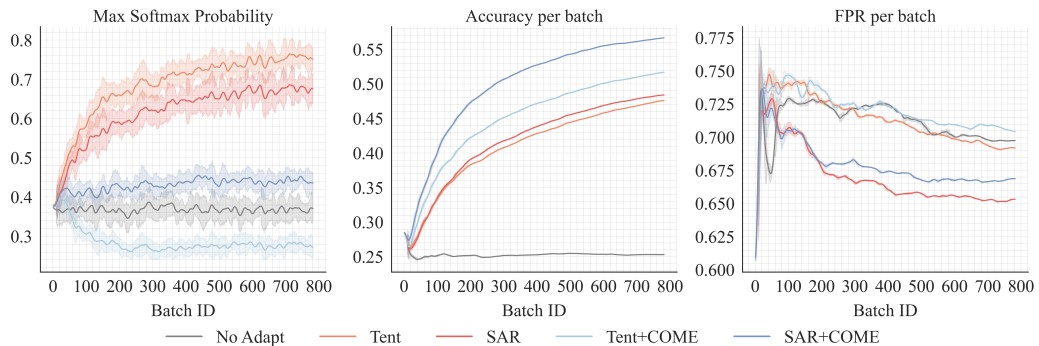

Figure 7: Comparison on two representative TTA methods on ImageNet-C under **Glass Blur** corruption of severity level 5. By contrast to EM, our COME establishes a stable TTA process with consistently improved classification accuracy and false positive rate.

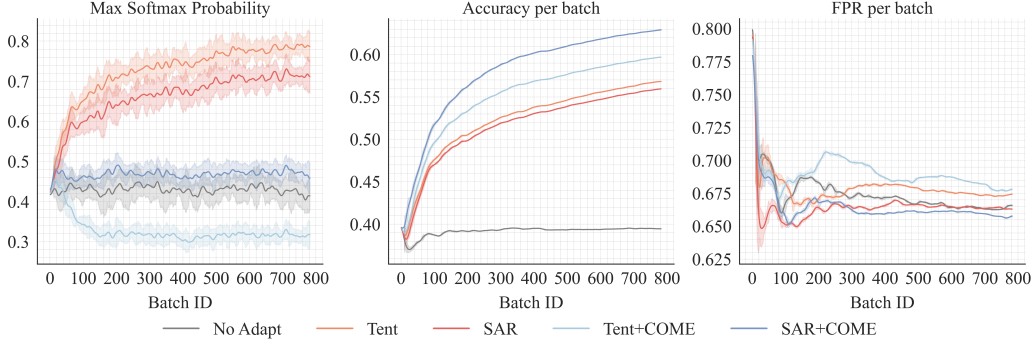

Figure 8: Comparison on two representative TTA methods on ImageNet-C under **Motion Blur** corruption of severity level 5. By contrast to EM, our COME establishes a stable TTA process with consistently improved classification accuracy and false positive rate.

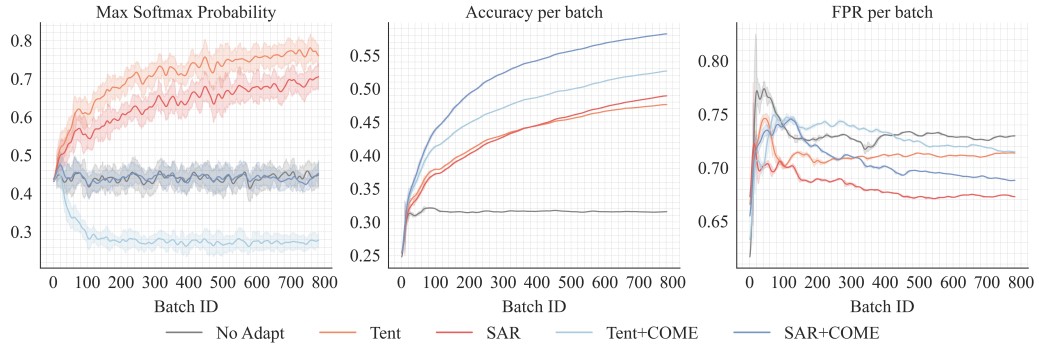

Figure 9: Comparison on two representative TTA methods on ImageNet-C under **Zoom Blur** corruption of severity level 5. By contrast to EM, our `COME` establishes a stable TTA process with consistently improved classification accuracy and false positive rate.

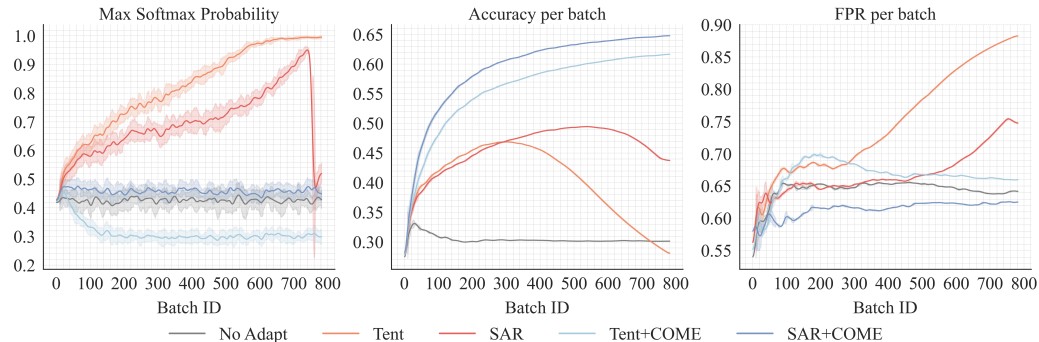

Figure 10: Comparison on two representative TTA methods on ImageNet-C under **Frost** corruption of severity level 5. By contrast to EM, our `COME` establishes a stable TTA process with consistently improved classification accuracy and false positive rate. Although the SAR method can recover the model when it collapses to a trivial solution, its performance remains poor. Our `COME` method addresses the issue of overconfidence that leads to model collapse.

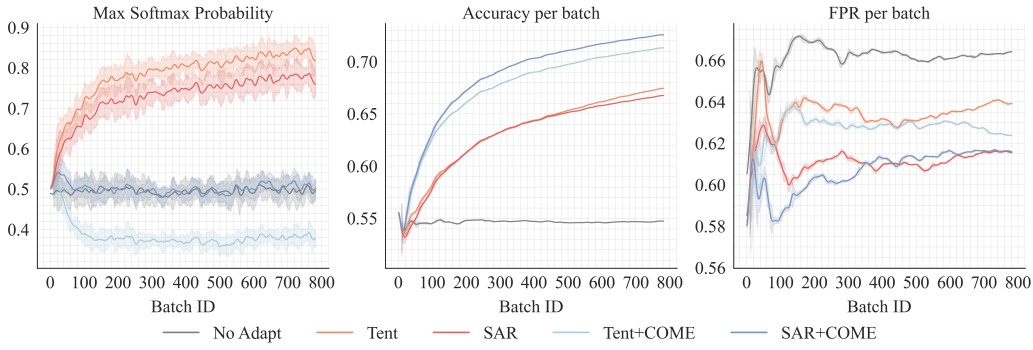

Figure 11: Comparison on two representative TTA methods on ImageNet-C under **Fog** corruption of severity level 5. By contrast to EM, our `COME` establishes a stable TTA process with consistently improved classification accuracy and false positive rate.

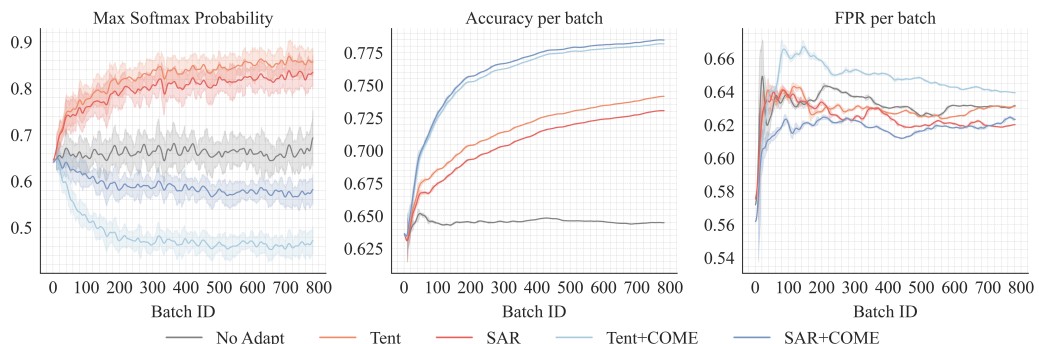

Figure 12: Comparison on two representative TTA methods on ImageNet-C under **Brightness** corruption of severity level 5. By contrast to EM, our `COME` establishes a stable TTA process with consistently improved classification accuracy and false positive rate.

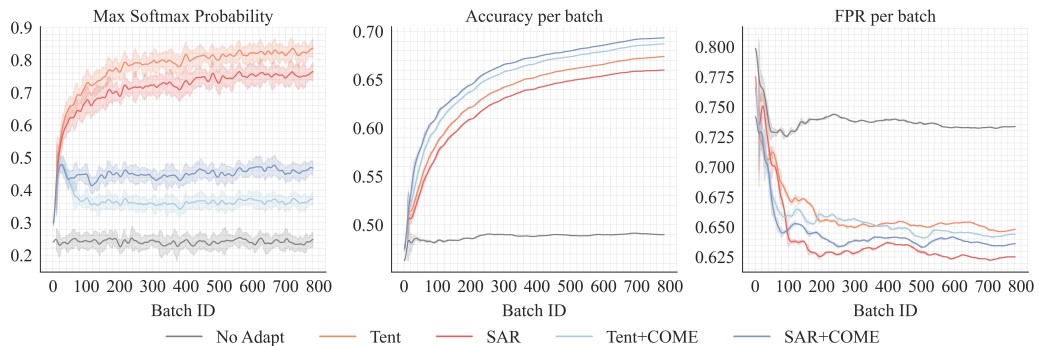

Figure 13: Comparison on two representative TTA methods on ImageNet-C under **Contrast** corruption of severity level 5. By contrast to EM, our `COME` establishes a stable TTA process with consistently improved classification accuracy and false positive rate.

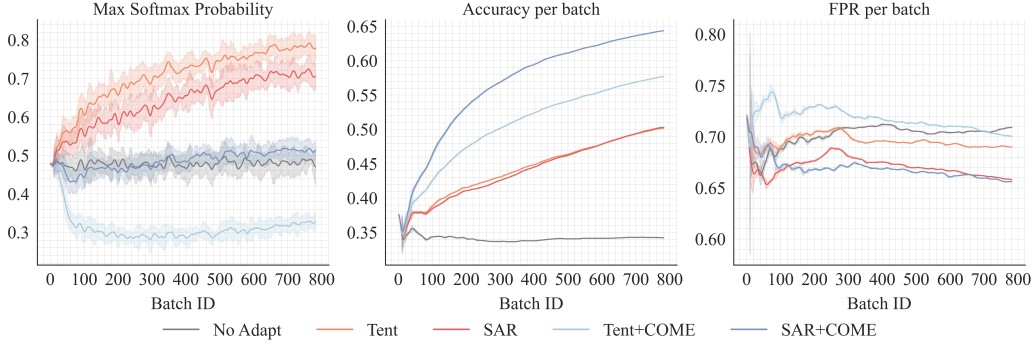

Figure 14: Comparison on two representative TTA methods on ImageNet-C under **Elastic Transform** corruption of severity level 5. By contrast to EM, our `COME` establishes a stable TTA process with consistently improved classification accuracy and false positive rate.

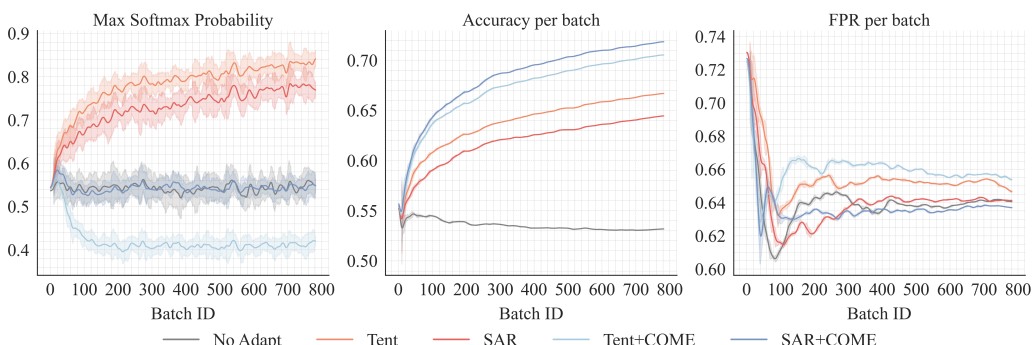

Figure 15: Comparison on two representative TTA methods on ImageNet-C under **Pixelate** corruption of severity level 5. By contrast to EM, our COME establishes a stable TTA process with consistently improved classification accuracy and false positive rate.

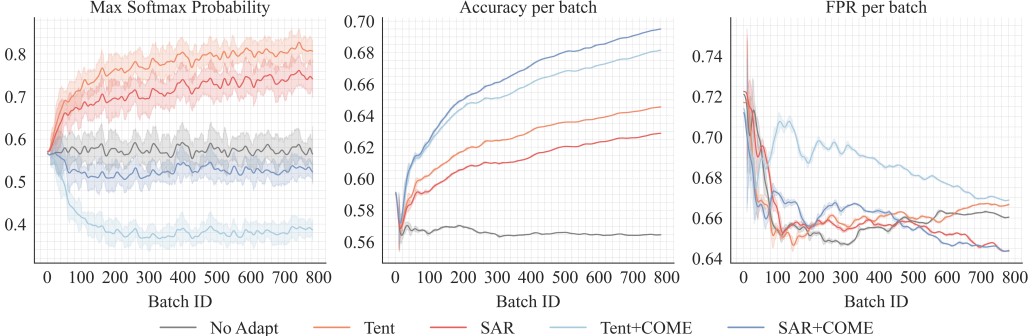

Figure 16: Comparison on two representative TTA methods on ImageNet-C under **Jpeg Compression** corruption of severity level 5. By contrast to EM, our COME establishes a stable TTA process with consistently improved classification accuracy and false positive rate.

Table 15: Additional results on ImageNet-S.

|  | Tent | EATA | SAR | COTTA |
|---|---|---|---|---|
| EM | $31.63 \pm 0.07$ | $36.11 \pm 0.19$ | $33.92 \pm 0.69$ | $30.84 \pm 0.35$ |
| COME | $39.22 \pm 0.04$ | $39.22 \pm 0.32$ | $43.52 \pm 0.52$ | $33.82 \pm 0.73$ |

Table 16: Comparison w.r.t. **in-distribution performance**, *i.e.*, on clean/original ImageNet validation set, with ViT as the base model. Substantial ($\geq 0.5$) improvement and degradation compared to the baseline are highlighted in blue or red respectively.

| COME | TENT | SAR | EATA | CoTTA | MEMO | Avg. |
|---|---|---|---|---|---|---|
|  | Acc↑ | Acc↑ | Acc ↑ | Acc ↑ | Acc↑ | Acc↑ |
| ✗ | 81.4 | 80.7 | 81.3 | 82.1 | 80.3 | 81.2 |
| ✓ | 83.1 | 83.1 | 83.1 | 82.8 | 80.6 | 82.6 |
| Improve | △1.7 | △2.5 | △1.8 | △0.7 | - | △1.4 |

# D  DISCUSSION

## D.1  ALTERNATIVE DESIGN CHOICE

**Choices of transformation function to obtain the opinion.** By definitions, the parameters of a Dirichlet distribution $\boldsymbol{\alpha}$ must be greater than 1 and the evidence $\boldsymbol{e}$ should be non-negative. This can be achieved by applying ReLU activation function or exponential function to the output logits as suggested in previous works (Han et al., 2022; Malinin & Gales, 2018). That is, we can get the evidence by

$$\boldsymbol{e} = \mathrm{ReLU}(f(x)) \tag{24}$$

or

$$\boldsymbol{e} = \exp f(x) - 1. \tag{25}$$

In this paper, we choose the exponential function. Since we assume the pretrain model is trained with standard cross-entropy loss, using exponential function to get the evidence can keep the training strategy unchanged. Besides, based on our early empirical findings, using exponential function can achieve better classification performance compared to ReLU.

We refer interested readers to (Malinin & Gales, 2018) and Gal's PhD Thesis (Gal et al., 2016) for more detailed implementation instructions and math deviations.

**Choices of uncertainty constraint.** In Lemma 1, we prove that by constraining on the model output logits, we can control the uncertainty mass $u$ not to diverge too far from the pretrained model. Previous work (Wei et al., 2022) proposes to mitigate the overconfidence issue by normalizing the logits during pretrain progress in supervised learning tasks. Following their implementation, we propose to optimize on the direction vector of $f(x)$, i.e., $f(x)/||f(x)||_p$, and thus we can expect that the optimization progress is not related to the magnitude of $f(x)$, i.e., its norm. Different from Wei et al. (2022), we recover the magnitude by multiplying the direction vector with its norm (detached), rather than a constant to avoid an additional hyperparameter. However, the uncertainty estimated by pretrain model may not be ideal. However, please kindly remind that in fully TTA task, we can only access unlabeled test data coming online and the inference efficiency matters. Thus traditional methods devised for handling overconfidence like calibration (Guo et al., 2017), ensembling (Zhou & Levine, 2021), BNNs (Huang et al., 2022) and other Bayesian methods like dropout (Gal & Ghahramani, 2016) are not applicable. The only practically available choice is to explore the uncertainty information contained within the model itself. As shown in previous works, while the softmax probability of pretrained model tend to be overconfident, subjective logic is much more reliable (Sensoy et al., 2018; Malinin & Gales, 2018), which can support the proposed regularization. Exploring more effective and efficient regularization is an interesting future research direction.

**Choices of $p$-norm.** The tightness for the upper and lower bounds in Lemma 1 is determined by the choice of $p$. By considering the simple model where $f(x)$ outputs the same logits for all classes, the ratio between the upper and lower bound is minimized by $p = \infty$. A larger p can lead to a more strict constraint on $|u - u\_0| \leq \delta$. We conduct additional experiments on varying $p$. When using

Table 17: Classification and uncertainty estimation comparisons under **open-world** TTA settings with **ResNet-50-BN**, where $P^{\text{test}} = 0.5P^{\text{Cov}} + 0.5P^{\text{Sem}}$(Gaussian noise of severity level 3) and a suit of diverse abnormal outliers as same with Table 3. Substantial ($\geq 0.5$) improvement and degradation compared to the baseline are highlighted in blue or red respectively.

| Method | COME | None Acc↑ | None FPR↓ | NINCO Acc↑ | NINCO FPR↓ | iNaturist Acc↑ | iNaturist FPR↓ | SSB-Hard Acc↑ | SSB-Hard FPR↓ | Texture Acc↑ | Texture FPR↓ | Places Acc↑ | Places FPR↓ | Avg. Acc↑ | Avg. FPR↓ |
|---|---|---|---|---|---|---|---|---|---|---|---|---|---|---|---|
| No Adapt | ✗ | 3.0 | 81.6 | 3.0 | 91.2 | 3.0 | 89.4 | 3.0 | 90.2 | 3.0 | 88.3 | 2.8 | 90.7 | 3.0 | 88.6 |
| PL | ✗ | 26.9 | 71.2 | 16.1 | 84.3 | 12.9 | 88.4 | 15.9 | 87.8 | 18.1 | 86.1 | 16.9 | 82.9 | 17.8 | 83.5 |
| TEA | ✗ | 28.5 | 73.5 | 17.6 | 84.0 | 9.6 | 84.5 | 11.8 | 88.3 | 19.9 | 84.2 | 16.8 | 82.9 | 17.4 | 82.9 |
| Tent | ✗ | 52.5 | 67.5 | 43.7 | 79.6 | 52.1 | 78.3 | 51.9 | 82.1 | 44.4 | 78.6 | 48.7 | 73.2 | 48.9 | 76.6 |
| | ✓ | 55.0 | 67.6 | 46.3 | 75.5 | 54.3 | 75.1 | 54.4 | 81.9 | 45.8 | 75.6 | 50.8 | 64.0 | 51.1 | 73.3 |
| | Improve | △2.6 | - | △2.6 | ▽4.2 | △2.2 | ▽3.2 | △2.5 | - | △1.4 | ▽3.1 | △2.2 | ▽9.2 | △2.2 | ▽3.3 |
| EATA | ✗ | 55.9 | 68.2 | 47.8 | 80.8 | 53.1 | 78.4 | 52.2 | 82.0 | 48.7 | 75.3 | 49.3 | 74.5 | 51.2 | 76.5 |
| | ✓ | 58.0 | 66.2 | 52.9 | 74.8 | 57.6 | 73.2 | 57.4 | 81.3 | 52.5 | 70.7 | 55.4 | 62.9 | 55.6 | 71.5 |
| | Improve | △2.0 | ▽2.0 | △5.1 | ▽6.0 | △4.5 | ▽5.1 | △5.2 | ▽0.7 | △3.9 | ▽4.6 | △6.0 | ▽11.6 | △4.5 | ▽5.0 |
| SAR | ✗ | 51.8 | 64.6 | 42.4 | 78.3 | 47.6 | 81.3 | 48.1 | 84.4 | 42.7 | 79.1 | 46.0 | 76.7 | 46.4 | 77.4 |
| | ✓ | 56.3 | 64.0 | 46.7 | 77.9 | 55.3 | 77.1 | 55.1 | 81.6 | 46.4 | 77.5 | 52.5 | 68.1 | 52.0 | 74.4 |
| | Improve | △4.5 | ▽0.6 | △4.3 | ▽0.3 | △7.7 | ▽4.2 | △6.9 | ▽2.8 | △3.7 | ▽1.6 | △6.5 | ▽8.6 | △5.6 | ▽3.0 |
| COTTA | ✗ | 22.6 | 70.7 | 14.4 | 87.4 | 21.1 | 78.6 | 19.7 | 84.1 | 15.5 | 87.3 | 15.8 | 82.0 | 18.2 | 81.7 |
| | ✓ | 24.5 | 69.4 | 14.7 | 86.1 | 21.4 | 81.6 | 19.4 | 86.1 | 16.0 | 85.7 | 16.4 | 82.6 | 18.7 | 81.9 |
| | Improve | △1.8 | ▽1.3 | - | ▽1.3 | - | △2.9 | - | △2.0 | △0.5 | ▽1.6 | △0.6 | △0.7 | △0.5 | - |
| MEMO | ✗ | 8.0 | 83.6 | 7.5 | 89.0 | 7.9 | 87.9 | 7.9 | 89.8 | 7.8 | 88.6 | 7.7 | 88.4 | 7.8 | 87.9 |
| | ✓ | 9.1 | 77.9 | 8.7 | 90.2 | 9.0 | 87.3 | 9.1 | 89.2 | 9.1 | 87.4 | 8.7 | 88.8 | 9.0 | 86.8 |
| | Improve | △1.1 | ▽5.7 | △1.2 | △1.2 | △1.2 | ▽0.7 | △1.2 | ▽0.6 | △1.3 | ▽1.2 | △1.1 | - | △1.2 | ▽1.1 |

Table 18: Comparisons w.r.t. computation complexity. Accuracy (%) and FPR (%) are average results on ImageNet-C (level 5) with ViT-Base. The Wall-Clock Time (seconds) and Memory Usage (MB) are measured for processing 50,000 images of ImageNet-C on a single RTX 4090 GPU.

| Method | COME | Acc ↑ | FPR ↓ | Memory | Run Time |
|---|---|---|---|---|---|
| No Adapt | ✗ | 39.8 | 67.5 | 853 | 59 |
| LAME | ✗ | 38.7 | 69.7 | 853 | 62 |
| T3A | ✗ | 41.0 | 67.7 | 984 | 179 |
| PL | ✗ | 51.3 | 69.1 | 6393 | 128 |
| FOA | ✗ | 53.7 | 63.6 | 869 | 1687 |
| TEA | ✗ | 46.9 | 68.3 | 17266 | 2865 |
| Tent | ✗ | 52.8 | 70.1 | 6393 | 129 |
| | ✓ | 61.2 | 66.5 | 6393 | 130 |
| | Improve | △8.4 | ▽3.6 | - | - |
| EATA | ✗ | 62.1 | 65.1 | 6394 | 135 |
| | ✓ | 64.5 | 63.8 | 6394 | 134 |
| | Improve | △2.4 | ▽1.3 | - | - |
| SAR | ✗ | 54.2 | 66.7 | 6393 | 253 |
| | ✓ | 64.2 | 63.8 | 6393 | 254 |
| | Improve | △10.1 | ▽2.9 | - | - |
| COTTA | ✗ | 46.1 | 67.9 | 19612 | 738 |
| | ✓ | 49.1 | 67.5 | 19611 | 739 |
| | Improve | △3.0 | ▽0.3 | - | - |
| MEMO | ✗ | 42.3 | 72.1 | 5392 | 20576 |
| | ✓ | 43.2 | 70.8 | 5392 | 20530 |
| | Improve | △0.9 | ▽1.3 | - | - |

Table 19: Comparison w.r.t. imbalanced label shifts performance. Results obtained on ViT and ImageNet-C (level 5) under **imbalanced label shifts** TTA setting, where the imbalance ratio is $\infty$. Substantial ($\geq 0.5$) improvement and degradation compared to the baseline are highlighted in blue or red respectively.

| Methods | COME | Noise Gauss. | Shot | Impul. | Blur Defoc | Glass | Motion | Zoom | Weather Snow | Frost | Fog | Brit. | Digital Contr. | Elast. | Pixel | JPEG | Avg. Acc↑ |
|---|---|---|---|---|---|---|---|---|---|---|---|---|---|---|---|---|---|
| No Adapt | ✗ | 35.1 | 32.2 | 35.9 | 31.4 | 25.3 | 39.4 | 31.6 | 24.5 | 30.1 | 54.7 | 64.5 | 49.0 | 34.2 | 53.2 | 56.5 | 39.8 |
| PL | ✗ | 49.7 | 48.6 | 50.9 | 49.8 | 41.5 | 53.0 | 41.9 | 26.6 | 49.0 | 64.3 | 73.6 | 65.6 | 45.2 | 63.9 | 63.0 | 52.4 |
| T3A | ✗ | 33.4 | 30.3 | 34.2 | 31.3 | 26.8 | 38.7 | 32.1 | 25.1 | 29.3 | 54.5 | 62.8 | 48.8 | 37.4 | 51.9 | 56.2 | 39.5 |
| TEA | ✗ | 44.9 | 40.3 | 46.3 | 39.8 | 35.2 | 46.0 | 12.1 | 14.3 | 46.9 | 60.3 | 72.7 | 60.2 | 48.6 | 62.7 | 58.8 | 45.9 |
| LAME | ✗ | 47.0 | 43.3 | 48.2 | 39.8 | 31.8 | 50.3 | 39.4 | 30.5 | 37.1 | 66.0 | 75.4 | 63.5 | 42.0 | 65.1 | 68.1 | 49.8 |
| FOA | ✗ | 41.5 | 39.2 | 43.6 | 42.5 | 33.7 | 45.5 | 41.0 | 44.9 | 44.5 | 60.1 | 67.7 | 58.8 | 45.7 | 57.3 | 62.7 | 48.6 |
| Tent | ✗ | 52.4 | 51.9 | 53.3 | 53.8 | 48.1 | 57.0 | 46.2 | 10.3 | 53.5 | 67.9 | 74.2 | 67.1 | 52.3 | 66.5 | 64.9 | 54.6 |
| | ✓ | 55.0 | 55.0 | 56.2 | 57.1 | 54.6 | 61.6 | 49.3 | 62.9 | 64.0 | 72.3 | 78.1 | 69.3 | 62.7 | 71.3 | 69.0 | 62.6 |
| | Improve | △2.5 | △3.2 | △2.9 | △3.4 | △6.5 | △4.6 | △3.1 | △52.6 | △10.6 | △4.4 | △4.0 | △2.2 | △10.4 | △4.9 | △4.1 | △7.9 |
| SAR | ✗ | 51.8 | 51.7 | 52.7 | 51.9 | 48.2 | 55.6 | 47.8 | 20.3 | 52.9 | 66.8 | 73.2 | 66.0 | 52.2 | 64.1 | 62.8 | 54.5 |
| | ✓ | 56.0 | 56.0 | 57.2 | 58.0 | 56.3 | 62.3 | 54.1 | 64.0 | 64.3 | 72.4 | 78.3 | 69.6 | 64.0 | 71.5 | 69.1 | 63.5 |
| | Improve | △4.2 | △4.4 | △4.5 | △6.2 | △8.1 | △6.6 | △6.3 | △43.8 | △11.4 | △5.7 | △5.1 | △3.6 | △11.8 | △7.4 | △6.2 | △9.0 |
| EATA | ✗ | 52.0 | 53.6 | 53.9 | 49.3 | 49.5 | 54.4 | 55.6 | 58.1 | 56.9 | 69.6 | 74.9 | 63.6 | 61.1 | 68.0 | 64.2 | 59.0 |
| | ✓ | 54.9 | 56.4 | 54.7 | 56.5 | 56.3 | 62.1 | 59.0 | 67.0 | 65.4 | 73.4 | 78.4 | 68.0 | 68.0 | 73.0 | 70.4 | 64.2 |
| | Improve | △2.8 | △2.8 | △0.8 | △7.3 | △6.8 | △7.7 | △3.4 | △8.9 | △8.5 | △3.9 | △3.5 | △4.4 | △6.8 | △5.0 | △6.1 | △5.3 |
| COTTA | ✗ | 42.9 | 40.0 | 44.6 | 36.0 | 29.7 | 44.8 | 37.2 | 42.3 | 46.4 | 60.7 | 72.9 | 65.0 | 45.4 | 61.6 | 62.9 | 48.8 |
| | ✓ | 51.6 | 49.0 | 52.9 | 41.7 | 37.0 | 51.6 | 43.8 | 46.7 | 53.2 | 65.9 | 74.4 | 65.6 | 52.8 | 66.7 | 65.9 | 54.6 |
| | Improve | △8.6 | △9.0 | △8.3 | △5.7 | △7.4 | △6.8 | △6.6 | △4.4 | △6.9 | △5.2 | △1.5 | △0.5 | △7.4 | △5.1 | △3.0 | △5.8 |
| MEMO | ✗ | 39.7 | 36.5 | 39.8 | 32.4 | 25.8 | 40.3 | 34.7 | 27.5 | 32.8 | 53.5 | 66.2 | 56.0 | 35.7 | 55.9 | 58.2 | 42.3 |
| | ✓ | 40.6 | 37.5 | 40.6 | 33.4 | 26.7 | 41.2 | 35.4 | 28.7 | 33.7 | 54.7 | 67.1 | 55.9 | 36.6 | 57.2 | 59.2 | 43.2 |
| | Improve | △0.8 | △1.0 | △0.8 | △1.0 | △0.9 | △1.0 | △0.7 | △1.2 | △0.9 | △1.2 | △0.8 | - | △0.9 | △1.3 | △1.1 | △0.9 |

Table 20: Comparison w.r.t. mixed shifts performance. Results obtained on ViT and ImageNet-C (level 5) under **mixed shifts** TTA setting, the performance is evaluated on a single data stream consisting of 15 mixed corruptions. Substantial ($\geq 0.5$) improvement and degradation compared to the baseline are highlighted in blue or red respectively.

| COME | TENT Acc↑ | FPR↓ | SAR Acc↑ | FPR↓ | EATA Acc↑ | FPR↓ | CoTTA Acc↑ | FPR↓ | Avg. Acc↑ | FPR↓ |
|---|---|---|---|---|---|---|---|---|---|---|
| ✗ | 58.0 | 72.3 | 53.6 | 68.2 | 58.8 | 71.3 | 62.0 | 69.7 | 58.1 | 70.4 |
| ✓ | 61.2 | 67.9 | 62.3 | 66.9 | 61.8 | 67.0 | 65.1 | 70.7 | 62.6 | 68.1 |
| Improve | △3.2 | ▽4.4 | △8.6 | ▽1.3 | △3.0 | ▽4.3 | △3.1 | △0.9 | △4.5 | ▽2.3 |

infinity norm, a suboptimal classification accuracy is observed. We suppose this is because an overly strict constraint can be harmful to TTA. Since on reliable test samples, we still expect to reduce the uncertainty (in a conservative manner).

## D.2 Limitations and Future work

Many state-of-the-art TTA methods are equipped with entropy minimization learning principle;, but the potential pitfalls lie in this optimization objective is not well understood. In this paper, we provide empirical analysis towards understanding the failure mode. These findings motivate us to further explore the connection between uncertainty learning and reliable TTA progress, which further implies a principle to design novel learning principle as an alternative to entropy minimization. Finally, we perform extensive experiments on multiple benchmarks to support our findings. In the work, a simple yet effective regularization on the uncertainty mass is devised, and other regularization techniques could be explored. Another interesting direction is further explore the relationship between overconfidence issue and model collapse theoretically.

