# OpenReview forum: "COME: Test-time Adaption by Conservatively Minimizing Entropy"
_ICLR.cc/2025/Conference — ICLR 2025 Poster_

### Official Review · Reviewer_5e1E · 2024-10-28

**Soundness:** 3
**Presentation:** 3
**Contribution:** 2
**Rating:** 6
**Confidence:** 3

**Summary:**

The paper proposes a Bayesian inference technique to address the overconfidence problem in test-time domain adaptation. Experiments demonstrate its effectiveness, achieving a SOTA performance.

**Strengths:**

- The motivation of the proposed method is clear.

- The idea of transforming the optimization problem with a constraint (Eq. (7)) into a simpler form (Eq. (9)) is interesting and effective, which significantly simplifies the problem.

- Several theoretical results, as well as empirical results, are provided. Theorem 1 helps quantitative understanding of the proposed method.

- Code is available.

**Weaknesses:**

- Experiments on the dependence of the results on $p$ and $\tau$ are lacking. I would like to see the results and discussions when $p \neq 2$ and $\tau \neq 1$.

- (Major) The paper applies a Bayesian inference (Eq. (5)) with a regularization to the overconfidence problem inherent in TTA. The "EM" (entropy minimization) mentioned in the paper is, in a more general context, the unsupervised learning using soft pseudo-label, which has a wide variety of applications beyond TTA.
While the proposed method is technically sound, it would benefit from discussion in a broader context—such as unsupervised learning with a pretrained model, unsupervised domain adaptation, source-free domain adaptation, and semi-supervised learning—to emphasize its wide applicability as a key contribution.

- (Major) Error bars are missing. Could you provide error bars because several performance gains are marginal?

- The performance metric "Avg." in the tables are nonsense, while it is actually a bad convention in the field. Obviously, a "1% gain" is quite different in iNaturalist and SSB-Hard, for example.

- (Minor) A large part of the paper is dedicated to reviewing previously known works, such as the overconfidence problem, which hinders readability.

- (Minor) Much of the paper, particularly the description of the proposed method, is redundant. A more direct tone is generally preferable in academic writing.

- Overall, while the paper is well-written and the proposed method is interesting and effective, the paper would benefit from a more refined articulation of its contributions and focus. Additionally, the reproducibility issue should be addressed.

**Questions:**

- (Eq. (6)) What if a hyperparameter $\lambda \in \mathbb{R}$ is introduced as $-\sum_{k=1}^K b_k \log b_k - \lambda u \log u$? This seems to be a straightforward generalization of Eq. (6).

- Is there any quantitative correspondence between $\tau$ and $\delta$ (Eq. (9) & (7))?

- How can we set $p$ and $\tau$ (or $\delta$) in practice?

- To my understanding, the regularizer (Eq. (9)) effectively prevents spurious training that would drive the second term in Eq. (6) to zero by enforcing $e \rightarrow 0$. Is this correct?

- To me, the proposed method seems to be a simple combination of known techniques (to clarify, I do *not* claim that simplicity  alone is grounds for rejection at all). Could you clarify the differences of the proposed technique from other Bayesian approaches, confidence calibration algorithms, semi-supervised learning, and entropy regularization techniques? A quantitative and objective discussion would be preferable, which would significantly enhance the paper's contribution and clarity.

---

> ### Author Response · Authors · 2024-11-20
> **Rebuttal 5e1E (1/2)**
>
> We appreciate your valuable and helpful comments on our paper and for recognizing the clear motivation and interesting idea. What you said makes sense and inspires us to think a lot.
>
> - Experiments on the dependence of hyperparameters are lacking. I would like to see the results and discussions when and discussions when $p\neq2$ and $\tau\neq 1$.
>
> Following your suggestion, we report the accuracy on ImageNet-C Gaussian noise level 5 with different hyperparameters. We implement our COME with Tent. The accuracy of the original Tent using entropy minimization is 52.6.
>
> |            | p=1  | p=2  | p=3  | p=$\infty$ |
> | ---------- | ---- | ---- | ---- | ---------- |
> | $\tau=0.5$ | 37.8 | 38.5 | 39.1 | 41.3       |
> | $\tau=1$   | 53.3 | 53.8 | 53.2 | 47.2       |
> | $\tau=1.2$ | 54.6 | 54.7 | 53.8 | 48.6       |
> | $\tau=1.5$ | 54.4 | 54.2 | 53.2 | 48.8       |
>
> Our COME generally outperforms EM with moderate hyperparamters.
>
> - While the proposed method is technically sound, it would benefit from discussion in a broader context—such as unsupervised learning with a pretrained model, unsupervised domain adaptation, source-free domain adaptation, and semi-supervised learning—to emphasize its wide applicability as a key contribution.
>
> We deeply appreciate this thoughtful and thorough suggestions. Here we discuss the settings you mentioned with additional experimental results.
>
> We agree that there is a strong connection between Source-Free Domain Adaptation (SFDA) and Test-Time Adaptation (TTA). TTA focuses on **online** adjusting during the testing. On the other hand, SFDA approaches generally perform **offline**. That is, the inference is deferred until the optimization is done. In contrast, our TTA method can achieve adaption and inference at the same time.
>
> To further validate the applicability of our method, we report the classification accuracy on ImageNet-C Gaussian noise level 5 under **source-free domain adpation settings**. The baselines we considered include pesudo label (PL), mutual information maxmization (IM), and entropy minimization (EM) following [1]. "-" means the model accuracy collapses to random guess level.
>
> | Epoch | PL    | EM   | IM    | COME (Tent) | COME (EATA) | COME (SAR) |      |
> | ----- | ----- | ---- | ----- | ----------- | ----------- | ---------- | ---- |
> | 1     | 31.55 | 0.55 | 60.71 | 66.50       | 68.38       | 66.83      |      |
> | 2     | -     | -    | 63.75 | 68.44       | 69.88       | 67.53      |      |
> | 3     | -     | -    | 65.07 | 69.20       | 70.07       | 68.06      |      |
>
> - Error bars are missing.
>
> According to your comments we provide the full results with standard deviation in **Appendix C.6** of the revised paper. Due to the large scale of ImageNet-C benchmark, the performance is stable with a relatively small standard deviation.
>
> - The performance metric "Avg." in the tables are nonsense, while it is actually a bad convention in the field. Obviously, a "1% gain" is quite different in iNaturalist and SSB-Hard, for example.
>
> Thanks you for this suggestion. We will avoid using such metric in the revision.
>
> - A large part of the paper is dedicated to reviewing previously known works, such as the overconfidence problem, which hinders readability. Much of the paper, particularly the description of the proposed method, is redundant. A more direct tone is generally preferable in academic writing.
>
> We appreciate your constructive comments on writing. We will carefully condense the introduction of related work and add more experimental results suggested by the reviewers instead.
>
> - What if a hyperparameter $\lambda$ is introduced as $-\sum_{k=1}^K b_k\log b_k-\lambda u\log u$? This seems to be a straightforward generalization of Eq. (6).
>
> This is an interesting idea. We believe that introducing this hyperparameter is practical. The magnitude of $\lambda$ reflects our confidence in whether the input sample should be classified into one of the known K class. To validate such an idea, we conduct additional experiments on the influence of different $\lambda$.
>
> Classification accuracy on ImageNet-C Gaussian noise level 5 with different $\lambda$.
>
> | $\lambda$ | 0.10  | 1.00 | 10.0  | 30.0  | 50.0  | 80.0  | 100  | 150   |
> | --------- | ----- | ---- | ----- | ----- | ----- | ----- | ---- | ----- |
> | Acc.      | 53.65 | 53.9 | 54.69 | 55.44 | 56.00 | 56.04 | 55.9 | 10.86 |
>
> We observe an additional performance improvement when $\lambda\in[1,100]$. We will add the results to the revised paper.

---

> ### Author Response · Authors · 2024-11-20
> **Rebuttal 5e1E (2/2)**
>
> - Is there any quantitative correspondence between $\tau$ and $\delta$ ?
>
> These two parameters play independent roles in constraining the uncertainty mass $u$. $\delta$ ensures that the uncertainty does not diverge too far from the pretrained model and thus avoids overly extreme output. Note that the uncertainty mass $u$ is negatively related to $\tau$ since we have $u=K/\sum \exp(f(x))$. That is, $\tau$ can increase or decrease the magnitude of $u$ before minimizing the entropy of opinion in Eq.6. For example, when $\tau$ is 1, $u$ averages to 0.3452 on ImageNet-C Gaussian noise level 5. And if we set $\tau=0.5$, $u$ averages to 0.4556, and at this time, the model will be more likely to reject to adapt most input samples.
>
> - How can we set $\tau$ and $\delta$ in practice?
>
> By introducing Lemma 1, we replace the hyperparameter $\delta$ with $p$. In our experiments, we set $\tau=1$ and $p=2$ for simplicity and in accordance with Occam's Razor. For $\tau$, as we mentioned before, it actually controls the magnitude of uncertainty mass. If we have some prior knowledge that most test samples should be rejected to adapt during TTA, we should choose a relatively small $\tau$ and making the model confidence more conservative in this circumstance. As for $\delta$ (or $p$) which represents the tolerance of uncertainty divergence, it should be selected per need by the user via trial and error: if users are extremely cautious about unreliable TTA, $\delta$ should be tuned down and $p$ should be tuned up; otherwise, if a better performance is required.
>
> - To my understanding, the regularizer (Eq. (9)) effectively prevents spurious training that would drive the second term in Eq. (6) to zero by enforcing. Is this correct?
>
> Yes. For unreliable samples of which the model outputs a relatively greater $u$, during entropy minimizing, the model will tend to increase $u$ and thus reject to train on such unreliable samples.
>
> - Could you clarify the differences of the proposed technique from other Bayesian approaches, confidence calibration algorithms, semi-supervised learning, and entropy regularization techniques? A quantitative and objective discussion would be preferable, which would significantly enhance the paper's contribution and clarity.
>
> Thanks for this valuable suggestion. (1) Compared to other Bayesian TTA methods like using Bayesian Neural Networks [2], or ensembling [3] which involves multiple models or inferences, the proposed COME does not need additional inference cost or modifying the model architecture which meets the demand of TTA. (2) Calibration methods typically rely on an individual validation set for tuning the temperature. However, during TTA, we can only access (batches of) unlabeled test samples coming online, thus calibration is not applicable. (3) Compared to semi-supervised learning like entropy regularization, our COME is a refinement of classic entropy minimization which is specified for online TTA settings. We provide a comprehensive comparison of these methods from the aspects of **application scenarios and empirical performance**.
>
> Comparison of the application scenarios of calibration, BNNs or ensemble based TTA, source-free domain adaption and COME on the dependency on labeled data, applicability for online TTA and additional inference latency.
>
> | Methods        | labeled data | online  | Inference latency |
> | -------------- | ------------ | ------- | ----------------- |
> | Calibration    | Yes          | No      | No                |
> | BNNs           | Yes          | No      | Yes               |
> | Ensemble       | No           | Yes     | Yes               |
> | Source-free DA | No           | No      | No                |
> | **COME**       | **No**       | **Yes** | **No**            |
>
> Classfication performance comparison with other bayesian insipired TTA methods, i.e., BACS, in **TTA setting**. The backbone is resnet50v2. Test data is ImageNet-C. The results of BACS [3] is copied from the original paper since it does not release the source code. Please kindly remind that BACS typically using an ensemble of 10 models thus is less efficient that our model-agnostic method.
>
> | No adapt | BN   | BACS | TENT | COME+Tent | COME+EATA |
> | -------- | ---- | ---- | ---- | --------- | --------- |
> | 47.3     | 47.6 | 56.1 | 48.9 | 51.1      | **58.2**  |
>
> Classification accuracy comparison with other learning objectives originate from semi-supervised learning, unsupervised learning in **TTA setting**. We implement all methods based on Tent and test on ImageNet-C Gaussian level 5.
>
> | No adapt | PL    | EM    | IM    | COME  |
> | -------- | ----- | ----- | ----- | ----- |
> | 35.12    | 49.85 | 52.39 | 50.98 | 53.77 |
>
> [1] Do We Really Need to Access the Source Data? Source Hypothesis Transfer for Unsupervised Domain Adaptation, ICML'20
>
> [2] Extrapolative Continuous-time Bayesian Neural Network for Fast Training-free Test-time Adaptation, NeurIPS'22
>
> [3] Bayesian Adaptation for Covariate Shift, NeurIPS'21

---

> > ### Comment · Reviewer_5e1E · 2024-11-21
> > **Reply**
> >
> > Thank you for your time and great effort. I have gone through other reviews, and the author's response addressed several concerns raised in my review: writing style, hyperparameter sensitivity, and relations of the proposed method to other models and tasks. I also acknowledge the additional experiments.
> >
> > Let me share some additional comments.
> >
> > - Error bars should be included in the additional experimental results provided in the author's response.
> > - Could you indicate where the additional experiments are integrated into the paper?
> > - For clarification, could you explain the source of the error bars (standard deviation)? Are they derived from random seeds for network initialization and mini-batches? I would appreciate the inclusion of additional details to support reproducibility.

---

> ### Author Response · Authors · 2024-11-21
>
> Thanks for your timely response. We have added the additional results to Appendix C. The newly intergrated results include:
>
> - Full results with standard deviation under standard TTA settings (Table 5 in Appendix C.1, page 17)
> - Additional comparison under source-free domain adaption settings (Table 7 in Appendix C.3, page 18)
> - New baselines suggested by the reviewers (Table 8 in Appendix C.4, page 18)
> - Additional results on varying hyperparameters (Table 9 and 10 in Appendix C.5, page 18-19)
>
> In the latest revision, we highlight the major revisions in blue.
>
> Due to space limit, we extract some results from Table 5 with standatd deviation here. The test data is ImageNet-C Gaussian noise level 5 (no adapt acc=35.1).
>
> |      | Tent         | SAR           | EATA          | COTTA         |
> | ---- | ------------ | ------------- | ------------- | ------------- |
> | EM   | $52.5\pm0.1$ | $51.9\pm 0.1$ | $56.0\pm0.2$  | $40.3\pm0.2$  |
> | COME | $53.9\pm0.0$ | $56.4\pm0.1$  | $56.1\pm 0.1$ | $43.5\pm 0.0$ |
>
> The source of the standard deviation consists 1) the order in which the test mini-batches coming online and 2) the randomness of the stochastic optimization algorithms, e.g., SGD, Adam. Since in TTA setting, the model is initialized from the publicly available pretrained model weights (i.e., via-base-patch16-224 from google and resnet50 from PyTorch), there is no randomness introduced by model initialization. All the experiments were conducted using random seed 2024, 2025 and 2026. We have added this explaination to Appendix (line 843-847) to enhance reproducibility.
>
> Thank you for considering our revisions and valuable suggestions. A newer manuscript is in preparation and we will actively working on this project. If you have any other concerns, please feel free to contact us and we look forward to disscuss with you.

---

> > ### Comment · Reviewer_5e1E · 2024-11-23
> >
> > Thank you for the reply. The authors addressed most of my concerns, and I changed my score from 3 to 5.
> >
> > > A newer manuscript is in preparation
> >
> > Have the authors revised the paper to resolve my remaining concern—namely, the presentation issue—which was also highlighted by other reviewers (and may impede an accurate evaluation of the paper)?

---

> > > ### Author Response · Authors · 2024-11-25
> > >
> > > Dear reviewer,
> > >
> > > We have uploaded the latest revision to improve the presentation quality. We understand the workload that reviewers face, and we appreciate your efforts already put into evaluating our work. If there are any additional insights, questions, or clarifications on our responses and manuscript that you would like to discuss with us, we would be very grateful to hear them, your feedback is valuable for the improvement of our research.
> > >
> > > Best regards,
> > >
> > > Authors

---

> > > > ### Comment · Reviewer_5e1E · 2024-11-26
> > > >
> > > > Thank you for sharing the revised paper. I have read it and confirmed the readability has been much improved. I changed my score from 5 to 6 because all the major and minor concerns raised in my first review have been resolved.
> > > >
> > > > I would like to share an additional comment below. Incorporating it into the paper would be impossible due to time limitation, but I hope they help.
> > > >
> > > > The current pretrained source models are ImageNet-pretrained ViT and ResNet. In view of recent trends in domain adaptation and related areas, I would like to recommend using rich foundation models, such as DINOv2 etc., which are expected to extract features that are much more robust against domain shifts than ImageNet-pretrained models. In real-world scenarios, using foundation models as the base architecture is more practical, to my understanding, because they potentially fill domain gaps without using any domain adaptation techniques. While foundation models are sometimes criticized for their large number of parameters, smaller, distilled models are sometimes available, as seen with DINOv2. In fact, a small version (ViT-S/14 distilled in  https://github.com/facebookresearch/dinov2) is as small as ResNet-50, addressing the latency problem. ImageNet-pretrained ViT and ResNet have been used as base architectures in this field, but they will become obsolete in near future, in my humble opinion.
> > > >
> > > > I wish you the best of luck with your work.

---

> > > > > ### Author Response · Authors · 2024-11-26
> > > > > **Thanks for your support!**
> > > > >
> > > > > Thank you for your support and insightful suggestions. I am happy to hear that all your concerns have been addressed. I will definitely explore foundation models in the future work. To me, generalization is one of the most fundamental and attractive challenge in machine learning. As a new in this field, I feel very lucky and motivated in discussion with you. Thanks again for your feedback and best regards.

---

> ### Author Response · Authors · 2024-11-23
>
> Thank you for your reply. Yes, we have uploaded a latest revision in which we make efforts in addressing the aforementioned presentation issues **just a few minutes ago before your newest comments**. Following your suggestions, we have
> - carefully condensed the introduction of related works and overconfidence issues (remove fig.2 and many sentences in Section 2 and 3 in the original manuscript)
> - provided a detailed explaination on design choices (line 209-220, 280-294)
> - illustrated possible generalization of the proposed method (line 235-238)
> - avoided using ``avg'' as a metric in the experiments (Table 1 and 2 in page 8)
> - clarified the source of standard deviation (line 843-847)
>
> We highlight the revised section in our manuscript in blue. We are deeply encouraged for your raising score. **It is our duty to address the presentation issues.** We will keep actively working on this. Your feedback is very valuable for us.

---

### Official Review · Reviewer_WHFP · 2024-11-01

**Soundness:** 3
**Presentation:** 3
**Contribution:** 2
**Rating:** 6
**Confidence:** 3

**Summary:**

This paper investigates the issue of model collapse in entropy minimization algorithms for test-time adaptation. The authors propose a novel entropy minimization approach that models prediction uncertainty by defining a Dirichlet prior distribution over model predictions. This method regularizes the model to favor conservative confidence for unreliable samples. Experiments on benchmark datasets demonstrate the effectiveness of the proposed algorithm.

**Strengths:**

The proposed algorithm introduces a rejection mechanism for unreliable samples in the TTA process, preventing the model from learning from potentially noisy labeled data. It is simple to integrate into existing TTA frameworks, and the experimental results indicate satisfactory performance.

**Weaknesses:**

The proposed method and its theoretical analysis rely heavily on existing techniques, which limits its technical novelty.

The core concept shares some similarity with research on learning with rejection. It is recommended to discuss how the proposed loss function compares with the loss functions used in learning with rejection, as outlined in [1].

There is a lack of experiments involving real-world applications with distribution shifts, as exemplified in [2]. Testing the proposed algorithm on real-world data streams in dynamic environments is suggested to validate its robustness.

References:

[1] Cortes, Corinna, Giulia DeSalvo, and Mehryar Mohri. "Learning with rejection." Algorithmic Learning Theory (2016).

[2] Yao, Huaxiu, et al. "Wild-time: A benchmark of in-the-wild distribution shift over time." Advances in Neural Information Processing Systems 35 (2022): 10309-10324.

**Questions:**

see weakness

---

> ### Author Response · Authors · 2024-11-20
> **Rebuttal WHFP**
>
> We thank the reviewer for the thoughtful and thorough comments on our paper.
>
> - The proposed method and its theoretical analysis rely heavily on existing techniques, which limits its technical novelty...It is recommended to discuss how the proposed loss function compares with the loss functions used in learning with rejection, as outlined in [1].
>
> Thanks for your comments. Here, we would like to discuss the differences between the proposed COME and the most related works including calibration and Bayesian uncertainty estimation methods. **Please kindly remind that in a fully TTA setting, we can only access unlabeled test samples coming online**. Calibration methods rely on an additional validation set which is not suitable for TTA task. Other Bayesian uncertainty estimation methods like ensembling, BNNs and dropout need multiple inferences or modifications on model architecture which is also unsuitable. Compared to most related subjective logic which needs labeled training data for superior uncertainty estimation. Our COME is specified for unsupervised online TTA settings. The idea of minimizing the entropy of opinion, the design of uncertainty constraint, and the analysis on model confidence are newly proposed in this work, which makes our method diverge far from existing works and staisfy the practical requirements of efficient and stable TTA.
>
> - Testing the proposed algorithm on real-world data streams in dynamic environments is suggested to validate its robustness.
>
> Thanks for your valuable suggestion. We agree that real-world data streams in dynamic environments is closely related to the proposed method. In table 3, we have provided the results under **lifelong** TTA settings where the test environments vary dynamically (e.g., **continuously varying** weathers like snow, frost, fog...). Such setting is also considered in recent TTA works specified for dynamic environments [2] [3]. We conduct additional experiments on the suggested benchmark [4] to further evaluate the proposed method. **Please kindly note that the official repository of the suggested wild-time benchmark is no longer actively maintained.** Instead, we introduce two more additional datasets, i.e., ImageNet-A and ImageNet-Sketch, to further evaluate the proposed method on commonly used OOD generalization benchmarks.
>
> Classfication accuracy on ImageNet-R (no adapt acc=35.15)
>
> |      | Tent  | EATA | SAR  | COTTA |
> | ---- | ----- | ---- | ---- | ----- |
> | EM   |   37.73    |    36.11  |  36.77    |   36.04    |
> | COME | 39.05 |   38.22   |   41.22   |   37.39    |
>
> Classification accuracy on ImageNet-S (no adapt acc=27.86)
>
> |      | Tent  | EATA | SAR  | COTTA |
> | ---- | ----- | ---- | ---- | ----- |
> | EM   | 31.63 |   39.26   |  33.92    |    30.84   |
> | COME | 39.22 |  41.85    |  43.52    |   33.82    |
>
> [1] Learning with rejection, ALT'16
>
> [2] Towards stable test-time adaptation in dynamic wild world. ICLR'23
>
> [3] Efficient test-time model adaptation without forgetting. ICML'22
>
> [4] Wild-Time: A Benchmark of in-the-Wild Distribution Shifts over Time. NIPS'22

---

> ### Author Response · Authors · 2024-11-25
> **Looking forward to your reply**
>
> Dear reviewer WHFP,
>
> We appreciate your efforts already put into evaluating our work. We posted rebuttal and a new revision including additional results on two large-scale benchmarks. As the discussion period is nearing its conclusion, could we kindly inquire if you have any remaining questions or concerns? Thanks for your valuable suggestions.
>
> Best regards,
>
> Authors

---

> ### Author Response · Authors · 2024-11-26
> **Additional results on dynamic environments (temporal distribution shifts)**
>
> Dear reviewer WHFP,
>
>
> Due to time limit, we have just completed the experiments results on yearbook from **WILD-time**. We agree with you that temporal distribution shifts is an interesting setting. **Since it is a very novel research direction, the benchmarks in this field are very limited and the repo of the original WILD-time is not actively maintained.** Thus we download the dataset from an unofficial third-part repo and also follow the very recent work [1] to simulate temporal distribution shifts. Specifically, we rotate the test image from ImageNet-R by a certain degree per batch. The results are as follows
>
> Classification accuracy on Yearbook (no adapt acc=84.5, backbone is a 6-layer fully convolutional network)
>
> |      | Tent  | EATA  | SAR   |
> | ---- | ----- | ----- | ----- |
> | EM   | 85.31 | 86.58 | 86.80 |
> | COME | 86.90 | 87.56 | 87.30 |
>
> Classification accuracy on ImageNet-R under temporally dynamic distribution shifts (no adapt acc=22.26, backbone is ViT-base)
>
> |      | Tent  | EATA  | SAR   |
> | ---- | ----- | ----- | ----- |
> | EM   | 24.44 | 23.45 | 24.48 |
> | COME | 27.21 | 25.63 | 28.90 |
>
>
>
> [1] Koebler, Alexander, et al. "Incremental Uncertainty-aware Performance Monitoring with Labeling Intervention." *NeurIPS 2024 Workshop on Bayesian Decision-making and Uncertainty*.

---

> > ### Comment · Reviewer_WHFP · 2024-11-26
> >
> > Thank you for your response and new experimental results. I believe my main concerns were addressed. As a result, I will increase my score from 5 to 6 accordingly.
> >
> > Here is the minor concern:
> > I mentioned that "The core concept of this paper shares some similarity with research on learning with rejection." By this, I meant that the use of a margin to reject predictions in [1] is conceptually similar to the approach proposed in this work. However, this work uses entropy to make decisions due to its unsupervised setting. It should be beneficial to discuss this line of research to make the paper more self-contained.

---

> > > ### Author Response · Authors · 2024-11-27
> > > **Thanks for your support**
> > >
> > > Thank you for your support and the additional suggestions regarding [1]. We will certainly discuss this strategy in our revision. If there are any further insights, clarifications, or questions you would like to share with us, please feel free to reach out. We truly appreciate your positive assessment and time in reviewing our paper.

---

### Official Review · Reviewer_4L2X · 2024-11-04

**Soundness:** 3
**Presentation:** 3
**Contribution:** 2
**Rating:** 5
**Confidence:** 5

**Summary:**

The authors propose Conservatively Minimizing Entropy, a method for test-time adaptation (TTA) that improves test-time adaption by managing prediction uncertainty. Unlike traditional entropy minimization, which can lead to overconfidence, COME uses a Dirichlet distribution to model uncertainty, allowing the model to avoid forced classification on unreliable data.

**Strengths:**

- This paper is well-motivated, and the story makes sense.
- Extensive experiments have been done to support the proposed method.

**Weaknesses:**

My major concerns include:
- For the proposed method: Why Dirichlet distribution is used? How is the Dirichlet distribution related to the final algorithm in Algorithm 1. In addition, what is the role of delta in Algorithm 1? It seems that the authors tell a long story about their algorithm, but the algorithm itself is rather simple.
- For the theoretical analysis: Could the authors provide a more detailed (theoretical) comparison between the proposed method and traditional EM? What is the benefit?
- For baselines: Some baselines are missing, for example,  [1] and [2].
- For the datasets: I'm curious why the authors follow literatures on outliers detection.
- For Theory 1: What is the exact benefit of the upper bound of model confidence? I think it will also hurt the performance on some "confident" samples.

[1] Nado, Z., Padhy, S., Sculley, D., D'Amour, A., Lakshminarayanan, B., & Snoek, J. (2020). Evaluating prediction-time batch normalization for robustness under covariate shift. arXiv preprint arXiv:2006.10963.
[2]Zhou, A., & Levine, S. (2021). Bayesian adaptation for covariate shift. Advances in neural information processing systems, 34, 914-927.

**Questions:**

Please refer to weaknesses.

---

> ### Author Response · Authors · 2024-11-20
> **Rebuttal 4L2X (1/2)**
>
> Thank you for valuable comments and recognizing of well-motivated paper and extensive experiments. Your comments are very helpful for us to improve the quality of our manuscript.
>
> - W1. Why Dirichlet distribution is used?
>
> In this work, we use Dirichlet distribution since it serves a prior distribution of categorical distribution in the Bayesian framework. Thus it is the most natural choice for modeling uncertainty of the predicted categoricals. There also exists several optional distributions, such as mixture of Dirichlet or the Logistic-Normal distribution. **We make such choice due to the tractable analytic properties of Dirichlet.** Besides, as we mentioned in lines 246-256, **compared to other Bayesian uncertainty modeling methods, using Dirichlet and SL for uncertainty modeling is model-agnostic and light-weight**, which meets the efficiency repuirements of TTA task.
>
> - W2.  What is the role of $\delta$ ?
>
> Thanks for your comments regarding $\delta$. As we mentioned in lines 281-285, **the role of $\delta$ is to constrain the uncertainty mass do not diverge too far away from the pretrain model in unsupervised TTA progress, which can prevent overly extreme model uncertainty**. The magnitude of $\delta$ represents our tolerance for uncertainty divergence. In lines 292-310, we propose to avoid to tune $\delta$ as a hyperparameter by Lemma 1. This significantly simplifies the problem, as recognized by Reviewer 5e1E.
>
> - Could the authors provide a more detailed (theoretical) comparison between the proposed method and traditional EM? What is the benefit?
>
> The proposed COME enjoys several advantages over EM. Please kindly remind that as mentioned in lines 265-269, compared to EM, COME allows the model to express high overall uncertainty and reject to adapt the test sample when the total evidence is insufficient. Besides, in lines 330-341, we provide a rigorous theoretical analysis on the model confidence. Our COME upper bounds model confidence during adaptation is based on the sample-specific confidence of the initial model. In contrast, the confidence of EM increases rapidly, which leading to model collapse as shown in Figure 1.
>
> - Some baselines are missing [1] [2].
>
> Thanks for your actionable suggestion. According to your advice, we conduct additional experiments involving the mentioned two baselines. Our COME outperforms them. Please kindly remind that **1) [2] has not made their source code publicly available and 2) most of our experiments are conducted on standard ViT thus [1] is not applicable since there is no batch normalize layers**. For a fair comparison, we ran the proposed COME using the same backbone (resnet50v2) and dataset (ImageNet-C) as in [2] and report the average accuracy. The results of [2] are copied from the original paper.
>
> | No adapt | BN   | BACS | TENT | COME+Tent | COME+EATA |
> | -------- | ---- | ---- | ---- | --------- | --------- |
> | 47.3     | 47.6 | 56.1 | 48.9 | 51.1      | **58.2**  |
>
> It is worth noting that [2] using an ensemble of **10 models** for TTA which introduces noticable inference latency. In constrast, our COME is desigined for **single** model TTA which is much more efficient.
>
> - I'm curious why the authors follow literatures on outliers detection.
>
> In this paper, besides standard TTA settings, we also consider open-world TTA tasks where there exists outliers in the test data. This setting is realistic and also considered in recent TTA works [3] [4] [5] as well as OOD generalization works [6] [7] [8]. We will clarify this in the revision to address your comments and avoid confusion.
>
> [1] Evaluating prediction-time batch normalization for robustness under covariate shift. arXiv preprint arXiv:2006.10963.
>
> [2] Bayesian adaptation for covariate shift. NIPS'21
>
> [3] Towards open-set test-time adaptation utilizing the wisdom of crowds in entropy minimization. CVPR'23
>
> [4] Unified Entropy Optimization for Open-Set Test-Time Adaptation. CVPR'24
>
> [5] ATTA: anomaly-aware test-time adaptation for out-of-distribution detection in segmentation. NIPS'23
>
> [6] Feed two birds with one scone: Exploiting wild data for both out-of-distribution generalization and detection. ICML'23
>
> [7] Realistic Unsupervised CLIP Fine-tuning with Universal Entropy Optimization. ICML'24
>
> [8] Unexplored Faces of Robustness and Out-of-Distribution: Covariate Shifts in Environment and Sensor Domains. CVPR'24

---

> ### Author Response · Authors · 2024-11-20
> **Rebuttal 4L2X (2/2)**
>
> - What is the exact benefit of the upper bound of model confidence? I think it will also hurt the performance on some "confident" samples.
>
> This is very insightful comments. As shown in Figure 1, EM leads to over-confidence and model collapse. Along the TTA progress, the model finally outputs nearly 100% confidence and collapses. In contrast, COME introduces a sample-wise upper bound on model confidence, which allows the model to get rid of such failure mode. **Theoretically**, we quantitively calculate the upper bound. For simplicity, we assume a Binary classification task and $\delta=0.1$. Typically we have $u_0\approx 0.2$ and $u_0\approx 0.6$ for normal test samples and anomaly outliers respectively. At this time, the upper bound of model confidence for normal test samples is 0.99995, and for anomaly outliers is 0.84113. We can observe that the upper bound for the normal test sample is still rather high. For this reason, we can suspect that such an upper bound would not hurt the performance of confident samples. **Empirically**, we recorded the **maximum model confidence** on test samples from ImageNet-C (1000 classes). When the prediction is correct, the confidence can still be very high (0.91), hence it does not harm performance. Besides, in contrast to entropy minimization, our COME established a **distinguishable** confidence margin between correct and wrong predictions. We will add these analysis in the revision to enhance the clarity of Theorem 1.
>
> |      | Right  | Wrong  |
> | ---- | ------ | ------ |
> | EM   | 0.9997 | 0.9978 |
> | COME | 0.9187 | 0.8186 |

---

> ### Author Response · Authors · 2024-11-25
>
> Dear reviewer 4L2X,
>
> We have posted rebuttal and a new revision including the suggested baselines and detailed clarifications. As the discussion period is ending soon, we kindly inquire if you have any remaining questions or concerns? Any insights, comments or questions that you would like to share on our manuscript is highly appreciated. Thanks for your efforts in reviewing our work and we sincerely looking forward to your reply.
>
> Best Regards,
>
> Authors

---

> > ### Comment · Reviewer_4L2X · 2024-11-25
> >
> > I would thank the authors for their reply. However, I still have some major concerns: (1) The overall approach is an approximation of Equation (6) only with some theoretical insights (Lemma 1), and there is no derivation for this. Further, the model confidence upper bound in Theorem 1 is based on the assumption of the original (precise) objective (Equation (6)). I would suggest the authors having better theoretical analysis on their approach. (2) I'm still concerned about the datasets used in this work. Why should we add outliers to test the test-time adaptation methods? From my perspective, we should pay more attention to different kinds of distribution shifts (like WILDS datasets typically used).
> >
> > Therefore, I would like to maintain my score.

---

> > > ### Author Response · Authors · 2024-12-01
> > > **A gentle reminder**
> > >
> > > Dear Reviewer 4L2X,
> > >
> > > Thanks for your feedback. In our response, we provided a detailed explanation for the choice of datasets and clarified that our method still outperforms its counterparts under 17 diverse distribution shifts in total (ImageNet-C, ImageNet-S and ImageNet-R) when there is no outlier. We also provided proof of Lemma 1 in Appendix A.
> > >
> > > We value your comments on our method. However, given that you are the only one who is negative about our paper, and the discussion period is ending soon, please feel free to let us know if you have any further comments on our work.
> > >
> > > Best regards,
> > >
> > > Authors

---

> > > > ### Author Response · Authors · 2024-12-02
> > > >
> > > > Dear Reviewer 4L2X, since Dec 2nd is the final deadline for public discussion, I would like to know if you have any concerns that we could address to improve the score. I also sincerely wish you great success with your own research.

---

> > > > > ### Comment · Reviewer_4L2X · 2024-12-03
> > > > >
> > > > > I would thank the authors for their response. However, my major concern still exists. I checked the proof and I did not mean there's no proof for Lemma 1. My major point is that, **there is no proof to show that Lemma 1 guarantees the constraints in Equation (6)**. Furthermore, in Theorem 1, the constraints in Equation (6) is directly assumed. Since your actual algorithm is Equation (8), there is a **mismatch** between them, which cannot justify or provide insights on your actual algorithm.
> > > > >
> > > > > Therefore, I would like to maintain my score.

---

> ### Author Response · Authors · 2024-11-25
>
> Thanks for your reply and new comments. We feel there exists significant misunderstanding and we would like to clarify
>
> - there is no derivation for this (Lemma 1)
>
> **In fact, we have provided the full proofs in Appendix A, line 717-724.**
>
> - the model confidence upper bound in Theorem 1 is based on the assumption of the original (precise) objective Equation. 6
>
> As we do have provided the proof of Lemma 1, this assumption is reasonable. Besides, as mentioned by other reviewers (5e1E and saue), Theorem 1 can aid to understand the proposed method.
>
> - Why should we add outliers to test the test-time adaptation methods?
>
> (1) Test-time adaption performs during the models' deployment in the real-world,  where it is evitable to encounter outliers. There have been **6 works [3-8] published on top-venue are of the same setting**. (2) **Open-world TTA is one of the three settings we considered.** We also conduct standard TTA experiments on ImageNet-C in Table 1. **Our method still outperforms its counterparts when there are no outliers**. (3) To further address the concerns on dataset, we conduct additional experiments on yearbook, a dataset from the suggested **WILD-time**. Since the official repo of WILD-time is not actively maintained, we manually download the dataset from this unofficial repo: https://github.com/wistuba/Wild-Time-Data/.
>
> Classification accuracy on Yearbook (no adapt acc=84.5, backbone is a 6-layer fully convolutional network)
>
> |      | Tent  | EATA  | SAR   |
> | ---- | ----- | ----- | ----- |
> | EM   | 85.31 | 86.58 | 86.80 |
> | COME | 86.90 | 87.56 | 87.30 |
>
> We promise to integrate the results into our latest revision. We are trying to conduct more experiments on other datasets from WILD-time like FMOW-time and the results will be integrated if time permit. Would you mind checking our responses and consider re-evaluating our manuscript? Thanks for your attention and best regards.

---

> ### Author Response · Authors · 2024-12-03
>
> Thanks for your comments. **It is our duty to correctly grasp your concerns and address them.**
>
> - there is no proof to show that Lemma 1 guarantees the constraints in Equation (6)
>
> As shown in Lemma 1, the uncertanty mass is constrained by the norm of evidence. Thus we can constrain on the norm instead
> of directly calculating the uncertainty mass and comparing them. That is, if the norm is unchanged, we can expect the uncertainty mass not diverge too far away. The logic here is natural, straightforward and recognized by other reviewers.
>
> - In Theorem 1, the constraints in Equation (6) is directly assumed...there is a mismatch between them (Eq.8 and Eq.6).
>
> We understand your point here and would like to argue that (1) Eq.8 is an **effective and practical approximation** of Eq.6, which do not strictly ensure exactly unchanged evidence norm due to the complexity of modern deep neural networks. Directly enforcing the invariance of the evidence's norm from Lemma 1 would lead to practical difficulties (inevitably requiring the storage of two copies of the model and explicit comparison). (2) **Such simplification can bring many practical benefits.** As mentioned by reviewer 5e1E, this is an **effective** and **interesting** design which significantly simplified the optimization. **Reviewer saue** mentioned that our method is **simple yet effective**, and the simplicity is its most **compelling** feature. (3) **Empirical study can support this design.** Please note that we provide empirical evidence that can support our theory in Figure 2, page 9 (the model confidence is much conservative as we expected). Our method achieves superior performance in practice. On large-scale ImageNet benchmarks under 15 diverse corruptions, our method substantially improve the average classification accuracy under standard TTA setting
>
> |             | Tent | EATA | SAR  | COTTA | MEMO |
> | ----------- | ---- | ---- | ---- | ----- | ---- |
> | EM          | 52.8 | 62.1 | 54.2 | 46.1  | 42.3 |
> | COME        | 61.2 | 64.5 | 64.2 | 49.1  | 43.2 |
> | Improvement | **8.4**  | **2.4**  | **10.1** | **3.0**   | **0.9**  |
>
> Thus we believe our method is reasonable, the explainary theory is helpful to aid understandanding (as mentioned by reviewer 5e1E), and the effectiveness of our design choice can be supported by extensive empirical observations.
>
> According to your comments, we will add this sentence below Eq.8 in our final revision to **make this point transparent**: `` Due to the complexity of modern deep neural networks, Eq.8 can not ensure exactly unchanged uncertainty mass. However, this approximation is effective and significantly simplifies the implementation. Empirical evidence supports this design can be found in...''. We greatly appreciate your open mind if you could further consider our clarification.

---

> ### Author Response · Authors · 2024-12-03
>
> Dear reviewer 4L2X,
>
> To further address your concern. We conduct additional experiments to investigate how the p-norm of evidence and uncertainty mass $u$ in our method change during TTA as additional empirical support to our theory. Here we show the mean value of 2-norm of evidence $b$ and uncertainty mass $u$ on ImageNet-C Gaussian noise. As we can see, our constraint makes the norm and $u$ to not diverge far from the pretrained model as we expected. This phenomenon can also support the assumption in Theorem 1.
>
> mean value of 2-norm of evidence
> | No Adapt | EM    | COME  |
> | -------- | ----- | ----- |
> | 52.35    | 57.23 | 51.46 |
> |     | +**4.88** | -**1.11** |
>
> mean value of uncertainty mass $u$
> | No Adapt | EM    | COME  |
> | -------- | ----- | ----- |
> | 0.1932    | 0.1159 | 0.2002 |
> |     | **-0.0764** | +**0.0070** |
>
> **Though we have no chance to discuss with you after your last comment, which was posted an hour before the discussion deadline, we hope our efforts can alleviate your concerns. We are frustrated to see that you raised your confidence level to an absolutely high 5 just an hour before the end of the discussion period after we have 1) provided comparisons with the suggested baselines 2) clarify the details of our method 3) provided empirical evidence to support the theory 4) evaluated our method on 17 diverse distribution shifts and cited 6 papers to explain the setting. At last, we sincerely thank you for the time and efforts you have put into our work and we authors will keep working to make it better.**
>
> Best regards,
>
> Authors

---

### Official Review · Reviewer_saue · 2024-11-05

**Soundness:** 3
**Presentation:** 4
**Contribution:** 3
**Rating:** 8
**Confidence:** 4

**Summary:**

This paper addresses the model collapse of the popular Entropy Minimization algorithm for Test-Time Adaptation. Motivated by the observation that the amplification of model over-confidence causes model collapse, this paper proposes to minimize entropy with respect to an augmented output distribution that includes the probability to reject a sample, which is an uncertainty estimation technique known as subjective logic. Moreover, a logit normalization is designed in order to avoid degenerated solutions. Theoretical analysis reveals that the resulting approach upper bounds model confidence during adaptation based on the sample-specific confidence of the initial model. The resulting algorithm, COME, can be easily embedded into general EM-based TTA methods with a few lines of code revision. Experiments across TTA, open-world and life long TTA settings demonstrate a significant and consistent improvement upon current EM baselines.

**Strengths:**

This paper accurately spots the paradox of EM's learning objective: minimization of entropy leads to over-confidence. And the paper proposes a simple yet effective solution to minimize entropy with respect to a probability distribution that faithfully estimates the uncertainty without over-confidence. It is a very reasonable idea to differentiate between the statistics used for prediction and for uncertainty estimation, which has long been considered the same in the TTA literature. Therefore, the algorithm enjoys the feature that the entropy minimization can be tailored to samples with different uncertainty, which is also supported by the monotonicity result in Theorem 1. The introduction of SL for uncertainty estimation is natural and perfectly compatible with softmax functions used for training models in most cases. As a result, the implementation is light-weight, model-agnostic, and extremely easy to embed into any TTA algorithms based on the EM objective. The experiments are convincing by covering both standard TTA tasks and more challenging settings of open-world TTA. A surprisingly significant 34.5% improvement on accuracy is reported on the model of SAR. And the algorithm has further addressed uncertainty estimation under continual distribution shift as a side product, which itself is also an important problem.

**Weaknesses:**

These are not necessarily weaknesses but rather some questions that I would like to confirm with the author.
1. How does the algorithm ensure that $b_k$ is non-negative for the computation of entropy, since $b_k$ is implemented as
$(e^{f_k(x)}-1)/ \sum_{k'} e^{f_k'(x)}$ which could be negative?

2. Why does the algorithm keep $u$ close to $u_0$? Does it imply that the uncertainty estimation for the pretrained model is trusted? What if the pretrained model is over-confident? What about the alternative constraint $u \geq u_0$ which seems to be more conservative as is the objective of COME?

3. Average false positive rate is used in experiments to assess uncertainty estimation. However, FPR measures the correctness of uncertainty estimation with such a binary perspective: for the samples we predict 1, what is the actual proportion of 0. Uncertainty estimation considers a more sophisticated question: for the samples we predict with a probability 0.7, is the actual proportion of 1 exactly 0.7? Expected calibration error (ECE) is a better metric in this sense.

4. Are there standard errors of the reported results?

**Questions:**

1. The tightness for the upper and lower bounds in Lemma 1 is determined by the choice of p. By considering the simple model where f(x) outputs the same logit for all classes, the ratio between the upper and lower bound is minimized by $p=\infty$. Is is a better choice to consider $\| f(x) \|_\infty = \max_k | f_k(x) |$?

2. The usage of exp transformation of logits in the softmax function appears pivotal to the proof of Theorem 1. And if we take b(x) as a general non-negative function of f(x), the upper bound may reduce to 1. And minimizing the entropy of opinion in equation 6 can also lead to the Dirac distribution, which is over-confident.  Is there a characterization for the class of functions that could be used to form a reasonable belief b(x)?

---

> ### Author Response · Authors · 2024-11-20
> **Rebuttal saue**
>
> Thanks for your thoughtful and thorough comments on our paper and for recognizing the contribution of our reasonable idea and the lightweight, model-agnostic, and extremely easy-to-implemented method.
>
> - W1. How to ensure $b_k$ is non-negative?
>
> Thanks for your careful reading. In our implementation, to ensure the non-negativity of $b_k$, we calculate the Dirichlet parameters by $\mathbf{\alpha}= \exp (ReLU(f(x))$. Note that $ReLU(f(x))$ is non-negative, and thus we have $\alpha\geq 1$ and $b_k=\alpha_k-1\geq 0$. This is to simplifies the proof of Theorem 1. Alternatively, we can also use $b_k=\exp(f(x))$ and $\alpha=b_k+1$.
>
> - W2. Why does the algorithm keep $u$ close to $u_0$? What if the pretrained model is over-confident?
>
> Very insightful question! We agree that the uncertainty estimated by pretrain model may not be ideal. However, please kindly remind that in fully TTA task, **we can only access unlabeled test data coming online and the inference efficiency matters**. Thus traditional methods devised for handling overconfidence like calibration, ensembling, and other Bayesian methods are not applicable. **The only practically available choice is to explore the uncertainty information contained within the model itself.** As shown in previous works [2] [3], while the softmax probability of pretrained model tend to be overconfident, subjective logic is much more reliable, which can support the proposed regularization. We promise to make this point transparent in the manuscript as a future research direction.
>
> - W2. What about the alternative constraint $u \geq u_0$?
>
> Certainly, constraint $u \geq u_0$ may result in a more conservative prediction. However, encouraging high uncertainty $u$ may result in underconfidence. That is, the model outputs all evidence equal to 0 since we have $u=K/\sum \exp f(x)$. During TTA, the model may encounter both unreliable samples and normal test samples that should be assigned high confidence. Unfortunately, due to the unsupervised fact of TTA, we lack of guidence to increase or decrease the uncertainty. Thus we directly constraint $|u-u_0|\leq \delta$ adhering to Occam's Razor.
>
> - W3. Expected calibration error (ECE) is a better metric in this sense.
>
> Following your suggestion, we conduct additional experiments and report ECE. A brief summary on ImageNet-C Gaussian noise level 5 are shown as follows and the fully results will be updated to the revised paper.
>
> |              | TENT  | SAR   | EATA  | COTTA |
> | ------------ | ----- | ----- | ----- | ----- |
> | Without COME | 26.61 | 19.22 | 21.44 | 19.43 |
> | With COME    | 13.79 | 17.89 | 19.01 | 16.31 |
>
>
>
> - W4. Are there standard errors of the reported results?
>
> According to your advice, we ran the experiments multiple times and report the standard errors. The results have been added to Appendix C in the revised paper.
>
> - Q1. The tightness for the upper and lower bounds in Lemma 1 is determined by the choice of p. By considering the simple model where f(x) outputs the same logit for all classes, the ratio between the upper and lower bound is minimized by $p=\infty$. Is is a better choice to consider $|f(x)|_{\infty}=\max |f_k(x)|$?
>
> We agree that a larger p can lead to a more strict constraint on $|u-u_0|\leq \delta$. We conduct additional experiments on varying $p$. When using infinity norm, a suboptimal classification accuracy is observed. We suppose this is because an overly strict constraint can be harmful to TTA. Since on reliable test samples, we still expect to reduce the uncertainty (in a conservative manner).
>
> | p=1  | p=2  | p=3  | p=$\infty$ |
> | ---- | ---- | ---- | ---------- |
> | 53.3 | 53.8 | 53.2 | 47.2       |
>
> - The usage of exp transformation of logits in the softmax function appears pivotal to the proof of Theorem 1. And if we take b(x) as a general non-negative function of f(x), the upper bound may reduce to 1. And minimizing the entropy of opinion in equation 6 can also lead to the Dirac distribution, which is over-confident. Is there a characterization for the class of functions that could be used to form a reasonable belief b(x)?
>
> Thanks for your questions. Here we discuss a few alternative functions to form $b(x)$. Common non-negative activation functions include **ReLU, softplus and exp**. For softplus function we can also derive a similar results as Theorem 1. However, if we calculate $b$ by $b(x)=ReLU(f(x))$, we can only derive a non-tirvial confidence upper bound (less that 1) when all the logits are non-negative. Since when the logits are all negative, $b(x)$ is an all zero vector and $u$ is a constant independent to the model confidence. Besides, we conduct experiments and observe that using exp transformation gains better performance than ReLU and softplus.
>
> | RELu  | SOFTPLUS | EXP   |
> | ----- | -------- | ----- |
> | 40.77 | 37.08    | 53.92 |
>
> [1] Evidential Deep Learning to Quantify Classification Uncertainty, NIPS'18
>
> [2] Predictive uncertainty estimation via prior networks, NIPS'18

---

> ### Comment · Reviewer_saue · 2024-11-25
>
> Thank the authors for their thorough response, and my gratitude also goes to the other reviewers for their efforts. I maintain my evaluation of this paper, which I believe makes a good contribution by applying uncertainty estimation theory and practices to address the well-known but significant challenge of overconfidence in TTA. The proposed approach is based on established yet highly relevant techniques from uncertainty estimation and calibrated learning. I identify the major contribution as connecting the fields of uncertainty estimation and TTA and solving the long-standing overconfidence problem of EM, with a universal and simple enough modification to each existing pipeline. I find the simplicity and yet effectiveness of the proposed method to be the most compelling feature.
>
> The rebuttal has addressed my concerns regarding W2, W3, and W4. In particular, I appreciate the standard error reporting, highlighting the significance of empirical results. Regarding W1, I suggest the authors clarify their implementation of the belief mechanism in the manuscript, and state any relevant simplifications to the theoretical model assumed for Theorem 1. For W2, I also suggest identifying the limitation of unresolved overconfidence in the pre-trained model.
>
> Overall, I maintain my positive evaluation of this paper and vote for its acceptance.

---

> > ### Author Response · Authors · 2024-11-25
> > **Thanks for your support**
> >
> > Thanks for your support and follow-up suggestions.
> >
> > We have carefully revised the manuscript following your latest suggestions, in which we detail the implementation of subjective opinion and theoretical assumptions. Besides, we identify the the limitations of current used regularization techniques in the conclusions: The simplicity of the uncertainty regularization used in our implementation is both an advantage and a limitation. On the one hand, constraining the uncertainty mass close to the pretrained model is easy-to-deploy and meets the efficiency requirements of TTA. On the other hand, this regularization may be less effective when the pre-trained model is also overconfident. We identify this as a limitation of our work. Exploring more effective regularization techniques for a better trade-off between the practical requirements of TTA and accurate uncertainty estimation could be a promising future direction.
> >
> > We feel very lucky to get such high-quality reviews and sincerely thank you and other reviewers for your professional suggestions. It's our duty to continue working on this project and ensure it meets the expectations of ICLR community.

---

### Author Response · Authors · 2024-11-24
**Looking forward to your reply**

Dear Reviewer, AC, SAC and PC,

We thanks all the reviewers for your valuable comments and for recognizing our contribution of reasonable and well-motivated idea (Reviewer saue, 4L2X), interesting design (Reviewer 5e1E), light-weight, model-agnostic, simple-to-integrate and extremely easy-to-implemented method (Reviewer saue, WHFP), extensive experiments with surprisingly significant, satisfactory improvement (Reviewer saue, 4L2X, WHFP), theoretical analysis that can aid understanding (Reviewer 5e1E). During rebuttal, we address the reviewers concerns with the following updates and improvements:

1. Full results with standard deviations (Table 5 in Appendix C.1, page 17).
2. New baselines, i.e., batch normalization (arxiv 2020) and using Bayesian ensemble for TTA (NeurIPS'21). The results are in Table 8, Appendix C.4, page 18.
3. Two new benchmarks, i.e., ImageNet-R and ImageNet-S instead of the suggested unactively maintained wild-time (Table 11, 12 in Appendix C.6, page 19). We appreciate the kindly understanding from the reviewer.
4. Discussion about the advantages of the proposed method beyond classic learning with rejection methods in unsupervised online TTA setting (line 210-220).
5. Math details and more clarification of the established theory (line 287-297).
6. Point-by-Point Responses: Comprehensive responses to specific reviewer comments.

We highlight them in **blue** in the latest revision. As the ICLR public discussion phase is ending soon, we kindly encourage you to share any feedback or questions on our submission while there’s still time. We’d be happy to address any concerns or provide clarifications to assist with your evaluation. We wish you success in your own research too.

---

### Meta-Review · Area_Chair_B2GA · 2024-12-18

**Metareview:**

This paper studies the issue of model collapse in entropy minimization algorithms for test-time adaptation. It proproses a entropy minimization approach that models prediction uncertainty by a Dirichlet prior distribution over model predictions. This method regularizes the model to favor conservative confidence for unreliable samples due to softmax. Experiments on typical benchmark datasets demonstrate the effectiveness of the proposed algorithm. I recommend to accept and suggest the authors to conduct one additional experiment that checks the approximation for Eq. (6). In the experiments, two-layer neural networks are sufficient to verify this if the constrained optimization is non-easy to be solve for deep neural networks.

**Additional Comments On Reviewer Discussion:**

After the rebuttal, most of the issues are handle. Only one issue left is that: no proof is given for Lemma 1 that guarantees the constraints in Eq. (6). I checked the authors' comments and understood the reviewer's concern. I suggest the author to add one additional experiment to compare the setting of using Eq. (6) and its approximation. In the experiments, two-layer neural networks are sufficient to verify this if the constrained optimization is non-easy to be solve for deep neural networks.

---

### Decision · Program_Chairs · 2025-01-22

Accept (Poster)